# Differential effects of sex and age on daily and infradian rhythms of mice

Pishan Chang[1], Timna Hitrec[1,2], Charlotte Muir[1], Meida Sofyana[1,3], Vuong Hung Truong[4], Shannon Lacey[1], Lukasz Chrobok[1], Jihwan Myung[4] and Hugh D. Piggins[1]

[1] *School of Physiology, Pharmacology & Neuroscience, University of Bristol, University Walk, Bristol, UK*
[2] *Department of Biomedical and Neuromotor Sciences, Università di Bologna, Bologna, Italy*
[3] *Department of Physiology, Faculty of Medicine, Public Health and Nursing, Universitas Gadjah Mada, Yogyakarta, Indonesia*
[4] *Graduate Institute of Mind Brain and Consciousness, Taipei Medical University, New Taipei City, Taiwan*

Handling Editors: Jing-Ning Zhu & Mark Viggars

The peer review history is available in the Supporting information section of this article (https://doi.org/10.1113/JP289676#support-information-section).

**The Journal of Physiology**

**Abstract figure legend** Summary of age- and sex-dependent differences in behavioural rhythmicity and environmental adaptation in mice. Middle-aged mice of both sexes show weight loss and increased daytime activity compared with young adults. In males, circadian timing remains precise with a strong 10-day infradian rhythm across ages. In young females, circadian and 5-day infradian rhythms interact to shape daily activity patterns, whereas in middle age, females display elongated and more precisely timed daily activity accompanied by a weakening of the 5-day rhythm.

**Abstract** Intrinsic biological rhythms regulate key physiological and behavioural processes, yet the influence of sex and age on these rhythms is not fully understood. We comprehensively examined 24 h (circadian) and >24 h (infradian; 5 and 10 day) rhythms in wheel-running and ingestive behaviours in single-housed young and middle-aged male and female mice. Circadian analysis revealed that middle-aged mice, particularly females, exhibited more precise daily rhythms and shifted a greater proportion of activity and feeding to the lights-on phase than young female mice. Middle-aged animals also ran for longer durations per day, suggesting age-related changes in activity regulation. Analysis of infradian rhythms further highlighted sex- and age-specific differences. Young female mice displayed robust 5 day rhythms in wheel-running activity, which were absent in middle-aged females. In contrast, few males (young or middle-aged) showed significant 5 day rhythms. Ten-day rhythms were most prominent in male mice, while females rarely expressed this periodicity. Physiologically, middle-aged mice lost more body weight in response to single housing, with middle-aged females being most affected. Interactions among behavioural rhythms in females also showed greater complexity, which increased with age. These findings reveal distinct sex- and age-dependent patterns in circadian and infradian rhythms as well as in physiological responses to isolation. Our work highlights the need to account for sex and age in chronobiological research, with broader implications for understanding vulnerability to age-related metabolic and behavioural disorders.

(Received 8 July 2025; accepted after revision 8 January 2026; first published online 5 February 2026)

**Corresponding authors** P. Chang and H. D. Piggins: School of Physiology, Pharmacology and Neuroscience, Biomedical Sciences Building, University of Bristol, University Walk, Bristol, BS8 1TD, UK.    Email: pi-shan.chang@bristol.ac.uk; hugh.piggins@bristol.ac.uk

**Key points**

- Physiological findings:

  - Middle-aged mice lost more body weight after single housing, with females most affected.

- Circadian findings:

  - Older mice show more daytime activity.
  - Precision in daily rhythm differs by sex and age.
  - Middle-aged females showed prolonged daily wheel running.

- Infradian findings:

  - Young females had robust 5 day rhythms, absent in middle-aged females.
  - Some males showed 5 day rhythms, but 10 day rhythms were most prominent in males.

- Complexity of rhythms:

  - Complexity of interactions among behavioural rhythms increases with age, particularly in females.

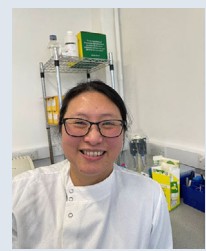

**Pishan Chang** completed her PhD in Neuroscience at University College London and is currently a Research Associate at the University of Bristol. Her research has focused on brain functions, including cognition and sensory processing, and now extends to the study of biological rhythms. Her current work focuses on elucidating how biological timing shapes vulnerability to ageing. Looking ahead, she aims to determine how rhythmic processes contribute to healthy ageing, how they influence age-related disease risk, and to develop strategies that may mitigate these effects.

## Introduction

Across all life forms, biological rhythms play an important role in the anticipation of, and adaptation to, cyclical environmental changes (Coskun et al., 2023). These rhythms can be classified based on their periodicity, including circadian (approximately 24 h) and infradian (>24 h) (Coskun et al., 2023; Golombek et al., 2014). In multicellular organisms, circadian rhythms are present in most cells and function to organise local physiology as well as to facilitate synchronisation of cellular and tissue processes (Green et al., 2008; Reppert & Weaver, 2002) with the external environment (Albrecht, 2012). In mammals, this is regulated by signals from the primary circadian pacemaker, located in the hypothalamic suprachiasmatic nucleus (SCN) of the brain. The SCN acts to orchestrate 24 h rhythms throughout the brain and body. As such, circadian rhythms play a crucial role in regulating daily cycles of sleep (Mattis & Sehgal, 2016; Meyer et al., 2022), activity (Hitrec et al., 2023; Hughes et al., 2021; Mendoza, 2007), hormone release (Koop & Oster, 2022) and metabolism (Asher & Schibler, 2011), and in optimally aligning these functions with environmental cycles (Hughes et al., 2021; Panda et al., 2002).

Infradian rhythms, defined as oscillations with periods longer than 24 h, also regulate a broad range of biological processes, including the menstrual cycle in humans (25–30 days) (Reed & Carr, 2000) and the oestrous cycle in rodents (4–5 days) (Byers et al., 2012; Cora et al., 2015). Unlike circadian rhythms, the mechanisms underpinning infradian rhythms are not localised to a particular structure and instead appear to arise from mechanisms distributed across multiple tissues and body systems (Hartsock et al., 2022). While circadian rhythms are well-characterized, emerging evidence suggests that infradian rhythms also shape behavioural and physiological outputs (Alvord et al., 2022). These slower oscillations may reflect endogenous cycles or responses to recurring environmental or social cues, such as metabolic demands, social interactions, or husbandry routines. Exploratory analyses of our long-term behavioural datasets suggested the presence of slower infradian rhythms (∼5–10 days), motivating us to investigate their prevalence, robustness and relation to sex and age.

Recently, there has been an increasing focus on sex as a biological variable, driven in part by the need to address a longstanding bias toward male-dominated studies in both preclinical and clinical research (Beery & Zucker, 2011; Zucker et al., 2022). Considering extensive sex differences in physiology and behaviour, the historical exclusion of females from research has likely had significant negative consequences for women's health (Beery & Zucker, 2011; Zucker et al., 2022). Further, evidence indicates sex differences in a range of biological rhythms such as the timing, amplitude and stability of circadian (Alvord & Pendergast, 2024; Pastrick et al., 2024) and infradian (Smarr et al., 2017) patterns of sleep–wake cycles (Goel et al., 2005; Kuljis et al., 2013; Mathew et al., 2019), hormonal fluctuations (Hatcher et al., 2020) and metabolic processes (de Souza et al., 2022; Qian et al., 2019).

Physical age is another factor that can significantly affect biological rhythms (Rahman et al., 2023). With advancing age, circadian rhythms often weaken, resulting in greater variability in rhythm precision, impaired synchronisation with environmental cues such as the light–dark cycle, and reduced internal synchronisation between physiological systems (Farajnia et al., 2012; Myung et al., 2023; Nakamura et al., 2011). Similarly, infradian rhythms may become irregular or decline and dampen over time (Ecochard, Leiva et al., 2024; Ohara et al., 2020; Takasu et al., 2015) and these changes can be sex-specific; for example, hormonal changes occurring during menopause (Brown & Gervais, 2020). Such age-related disruptions to biological rhythms can impair physiological homeostasis, contribute to sleep disorders, and accelerate the progression of age-related pathological processes (Mattis & Sehgal, 2016).

While the role of biological rhythms in synchronising internal processes with external cues is well-established, the influence of sex and age on these rhythms and their interactions remains unclear. In this study, we monitored daily (circadian) and infradian rhythms in wheel-running, drinking and eating activities. We investigated how these, as well as body weight, differed between young (3–7 months) and middle-aged (8–16 months) male and female mice in single-housing conditions. For infradian rhythms, we chose to study 5 and 10 day rhythms, since these are the periodicities of the intrinsic (the female oestrous cycle) (Byers et al., 2012; Cora et al., 2015) and extrinsic regulation, such as sensitivity to the cage-changing schedule as a periodic environmental perturbation, respectively.

We report age- and sex-related differences in both daily and infradian wheel-running rhythms (and to a lesser extent, ingestive behaviour rhythms), with interactions between different rhythms increasing in complexity with age in female mice. The strength of daily rhythms in wheel running increased with age in female but not male mice. Rhythms of 5 day periodicity in ingestive behaviour and wheel running declined with age in female mice but were not detected in male mice. Wheel-running rhythms of 10 day periodicity were found in young and middle-aged male mice, but not significantly in female mice. Further, younger female mice, which exhibited strong 5 day behavioural rhythms, showed less weight loss during the initial period of single housing. In contrast, middle-aged females displayed weaker 5 day rhythms and experienced greater weight loss, suggesting

that age-related weakening of infradian rhythmicity may contribute to reduced physiological resilience. By contrast, stronger daily wheel-running rhythms were associated with reduced body-weight loss in male mice but increased weight loss in female mice. These findings highlight the importance of monitoring multiple behavioural rhythms as well as the inclusion of male and female mice in the study of ageing. Gaining a deeper understanding of the dynamics in these rhythms is important for advancing targeted strategies to address age-related and sex-specific vulnerabilities. It may ultimately inform personalised approaches in chronobiology.

## Materials and methods

### Ethical approval

All experiments complied with the United Kingdom Animals (Scientific Procedures) Act 1986 and are reported following the ARRIVE Guidelines for Reporting Animal Research developed by the National Centre for Replacement, Refinement, and Reduction of Animals in Research (NC3Rs), London. All procedures were performed under a UK Home Office Project Licence and received approval from the local Animal Welfare and Ethical Review Body at the University of Bristol (ASU289). At the end of the experiment, animals were humanely killed by an intraperitoneal overdose of pentobarbital (200 mg/kg), in accordance with Schedule 1 of the UK Animals (Scientific Procedures) Act 1986. Death was confirmed by the absence of vital signs, including cessation of heartbeat and respiration, and lack of reflex responses, followed by cervical dislocation as a secondary method to ensure death.

### Animal

Experiments were conducted on young (3–7 months old) and middle-aged (8–16 months old) male ($n = 33$) and female ($n = 38$) (Arellano et al., 2024; Flurkey et al., 2007) PER2::LUC mice (on a C57BL/6 background) (Yoo et al., 2004). The age range of 8–16 months in mice is commonly considered middle-aged in the context of neuroscience and ageing research (Arellano et al., 2024; Flurkey et al., 2007). According to lifespan estimates in C57BL/6J mice, this period corresponds approximately to 30–50 human years (Jackson et al., 2017), a phase often marked by early signs of physiological ageing but before the onset of advanced decline (Shoji et al., 2016). This age window is particularly relevant for studying age-related changes in sleep, circadian rhythms and behaviour, as subtle differences begin to emerge without the confounds of severe deterioration seen in old age (Shoji et al., 2016).

All experimental animals were homozygous PER2::LUC mice, and comparisons were made within this genotype effects. The advantage of using PER2::LUC mice is that they retain a functional molecular circadian clock, enabling follow-up studies on tissue-specific rhythmicity without altering the animal model. This approach ensures consistency across behavioural and molecular datasets in longitudinal research. Initial breeding stock was supplied by Pat Nolan (MRC Harwell Institute, UK) and Michael Hastings (MRC Laboratory of Molecular Biology, UK). Subsequently, breeding of these mice was conducted by a specialised team at the University of Bristol under standardized conditions to ensure animal welfare and experimental consistency. The light cycle in the breeding rooms was lights-on from 05.00 h to 19.00 h. Adult mice were requested as needed and transferred from group-housing conditions in the breeding unit to the experimental unit. At the start of the experiment, body weight was recorded every 10 days during single housing. Because animals were assigned to age-based cohorts, not all mice entered single housing at the same chronological age. Nonetheless, weight changes were consistently evaluated over the same 10 day period across all groups to ensure comparability.

Throughout the experimental period, animals were maintained on a standard laboratory diet containing a minimum of 22.0% crude protein, 3.5% crude fat, a maximum of 6% crude fibre, and a maximum of 12% moisture (LabDiet EURodent, PMI Nutrition International, LLC). During the experiment, the animals were housed individually in polycarbonate cages equipped with running wheels under controlled environmental conditions (21–22°C, 50–60% humidity, 12 h light:12 h dark; LD cycle) and had free access to food and water. The average light intensity during the light phase was ∼289.6 lux. Two combinations of cages and running wheels were used: large-cage setups (43 × 27 × 15 cm; length × width × height) with stainless steel running wheels (12 cm diameter), and small-cage setups (32.5 × 14 × 15 cm) with stainless steel running wheels (11 cm diameter).

### Data collection

Animals used in these studies were individually housed in cages equipped with running wheels, following protocols established in our laboratory (Hitrec et al., 2023; Hughes et al., 2021). Activity data were recorded in the large cages using the PhenoMaster acquisition system (10 min intervals, TSE Systems, Germany) and in the smaller cages using ClockLab (30 s intervals, Actimetrics, USA). Feeding and drinking behaviours were also monitored and recorded in the larger cages using the TSE system (each cage is equipped with weighing sensors which record how water and food changes over time). At the start of

behavioural screening, mice were maintained in a 12:12 light–dark cycle (LD12:12) for 30–40 days. Subsequently, for some animals ($n = 52$), the lights were turned off, and the animals were kept in constant darkness (DD) for 10 days and then they were returned to LD conditions for the remainder of the experiment.

## Data analysis

**Determination of circadian and infradian periodicity.** In this study, the Lomb-Scargle (LS) periodogram, as implemented in ActogramJ (Schmid et al., 2011) (available at https://bene51.github.io/ActogramJ/index.html), was used to analyse circadian and infradian periodicity, including the detection of peak periodicities and the strength of rhythmic activity. The LS periodogram is a well-established algorithm designed to detect and characterize periodic signals in both evenly and unevenly sampled data (Ruf, 1999; Tackenberg & Hughey, 2021; Van Dongen et al., 1999). Importantly, LS does not require continuous sampling, and short gaps in the data do not bias the estimated period.

For daily/circadian analysis, 10 consecutive days of data collected under LD and/or DD conditions were used to estimate daily and circadian activity, respectively. The frequency range was set between 0.6 and 1.4 days, corresponding to rhythms close to a 1 day (24 h) cycle. For each animal, the peak frequency of the power spectrum was identified, along with the normalised power of the peak, which represents the strength of the circadian rhythm.

For infradian analysis, 30 consecutive days of data collected under LD conditions were analysed to estimate rhythms with longer periodicities. The frequency range was defined as between 3 and 20 days to capture infradian cycles within this periodicity. The analysis identified the dominant frequency within this range, representing the primary infradian rhythm, and calculated the corresponding power, reflecting the amplitude or strength of this rhythm. Higher power values indicate a more pronounced rhythmic signal at a given frequency.

**Calculating activity on-/offsets.** ActogramJ was used to determine the onset and offset of daily/circadian activity. To prepare the data for analysis, the original activity data were smoothed using a Gaussian distribution with a standard deviation of 100 s to reduce noise and highlight rhythmic patterns. The smoothed data were then thresholded to identify periods of activity and inactivity. Specifically, the median of all non-negative activity values was calculated and used as the activity threshold. This approach ensured a robust baseline for distinguishing between active and inactive phases, allowing precise identification of daily activity onset and offset.

**Sliding-window Lomb-Scargle spectrogram.** To visualize the evolution and stability of infradian rhythms, we applied sequential windows of data from the activity timeseries to generate a time-resolved spectrogram of periodicity (Hughes et al., 2015). LS spectrogram with a sliding window was performed using custom Python scripts based on the astropy.timeseries.LombScargle module. This method estimates rhythm strength across defined period ranges. LS periodograms were calculated within sliding windows of 3 days (circadian), 11 days (infradian, ~5 days) or 21 days (infradian, ~10 day), each advanced in 1 day increments across the recording period. For each window and frequency range, the normalised LS power was plotted as a heatmap. This approach enabled the visualization of both persistent and transient rhythms throughout the full experimental duration. While the full-range LS periodogram identifies dominant infradian rhythms, the sliding-window spectrogram revealed that such rhythms were not always stable over time.

## Daily activity patterns analysis

Diurnal pattern analysis was carried out in the Circa Diem MATLAB toolbox (Mathworks, Natick, MA, USA) developed by Dr Joram J. van Rheede, University of Oxford, UK (v0.1, available on https://github.com/joramvanrheede/circa_diem). Once the data was imported (10 day intervals), the time of the date fit was calculated. In brief, data points were aligned to the time of day, referring to specific clock times within the 24 h cycle, by averaging values within defined time intervals. This approach allows for the visualization of daily trends, independent of the actual date, by focusing on patterns that repeat over each 24 h period. The input included time stamps (in MATLAB datetime format) and their associated values. Circular plots, known as rose plots, were then generated to display average data values at different times of the day over 24 h. The 24 h cycle was divided into equal segments, each representing a specific time interval (1 h). The height of each segment corresponded to the median value of the data points that fell within that time interval.

To provide further insight into the temporal structure of daily rhythms, the summarised patterns in data recorded over the 24 h day by treating time as points on a circle were applied. Each data point contributes information about the time of day (its direction (angle) on the circle) and how strong or significant that point was (its length). By combining all these points, a single summary vector was calculated for each animal. The direction of this vector showed the time of day when most activity occurs, while its length indicates how strongly the activity is concentrated. A longer vector means the activity is focused on a specific time, while a shorter vector means the activity

is more spread out. For the group comparison related to circular (i.e. direction) data, the Watson–Williams test (Circular Statistics Toolbox available on https://github.com/circstat/circstat-matlab) (Berens, 2009; Landler et al., 2021) was used. Differences were considered statistically significant at $P < 0.05$.

## Vaginal cytology method

To collect cells from the mouse vaginal canal, approximately 20 µl of saline was drawn into a pipette. The tip of the pipette was gently inserted into the vaginal orifice at a depth of approximately 1–2 mm and then the saline flushed in and out of the vagina and two or three times. The collected cells were transferred onto a clean, dry glass slide, air-dried, and then stained with 1% cresyl violet for 5 min. After staining, the slides were rinsed with water and examined under brightfield illumination at 20× magnification (Leica DM 4000B, Leica Microsystems UK Ltd). The stage of the oestrous cycle was determined based on the relative presence of leukocytes, cornified epithelial cells and nucleated epithelial cells.

In particular, we classified the stage based on well-established criteria. During pro-oestrus, the smear predominantly showed nucleated epithelial cells with some cornified epithelial cells; a few leukocytes are potentially present in early pro-oestrus. In oestrus, the smear was mainly composed of cornified epithelial cells. Metoestrus, a brief transitional stage, was marked by a mixture of cornified epithelial cells and poly-morphonuclear leukocytes, along with some nucleated epithelial cells in late metoestrus, reflecting the sloughing of the uterine lining. Dioestrus, the longest stage (lasting more than 2 days), was characterized primarily by poly-morphonuclear leukocytes, with occasional epithelial cells appearing in late dioestrus. During this phase, leukocytes dominated as they clear cellular debris (Byers et al., 2012).

## Statistical analysis

Detailed statistical analysis was performed using estimation statistics (Ho et al., 2019) (open source estimation program available on https://www.estimationstats.com), SPSS 28.0.0.0 (Statistical Product and Service Solutions, IBM) and MATLAB 2021a. All data are presented as medians with 95% confidence intervals (CIs), unless otherwise specified. Estimation statistics report mean differences (effect size) with expressions of uncertainty (CI estimates). In this method, each paired mean difference is plotted as a bootstrap sampling distribution, using 5000 bootstrap samples and the confidence intervals are bias-corrected and accelerated. The $P$ value(s) reported are the likelihood(s) of observing the effect size(s) if the null hypothesis of zero difference

is true. For each permutation $P$ value, 5000 reshuffles of the control and test labels were performed; $P < 0.05$ is considered a significant difference. The significance threshold for all correlation tests was set at $P < 0.05$. The additional comparisons of means were performed using two-factor ANOVA with Bonferroni's *post hoc* test or two-way ANOVA with Holm's *post hoc* test, where appropriate, if the data were normally distributed; the Kruskal-Wallis test with *post hoc* Dunn's multiple comparisons test if the data were not normally distributed (with the Shapiro–Wilk test used to assess normality of the data distributions).

Pearson's correlation coefficient ($R$) and linear regression were used to assess relationships between circadian and infradian rhythm parameters, weight change and the exact age (in months) of each individual animal. Age was treated as a continuous variable rather than only as the categorical grouping (young: 3–7 months; middle-aged: 8–16 months), as this approach captures within-group variability beyond categorical age classifications. Statistical significance was set at $P < 0.05$. The MATLAB function compare_correlation_coefficients by Sisi Ma (available at MathWorks File Exchange_44 658) was used to compare correlation coefficients between groups.

## Results

### Sex- and age-dependent changes in behavioural rhythms

**Wheel running.** We used wheel running as a measure of daily voluntary exercise to assess daily rhythms in locomotor activity. Two types of cage environments were used, large and small cages (see *Methods*), with ingestive behaviour (food and water consumption) also monitored in the large cages (the small cages were not equipped with devices to measure these parameters). Independent of sex and age, mice housed in large cages and maintained under 12 h:12 h light–dark (LD) conditions exhibited the expected nocturnal pattern of wheel running. Assessment of the period of this daily rhythm revealed a ∼24 h period across both sexes and age groups, with no significant differences observed in young or middle-aged male and female mice (Fig. 1*A–C*; Table 1). Environmental lighting can influence the period and strength of daily rhythms (Hughes et al., 2021) and to assess intrinsic circadian rhythms in wheel running, we transferred the mice to constant dark (DD) conditions for 10 days. The circadian free-running period for each age/sex group remained close to 24 h and did not differ between the groups (Fig. 1*A–D*; Table 1). However, the strength of the rhythm declined in middle-aged female mice compared with middle-aged male mice, as indicated by the reduction in spectral power in DD relative to LD (DD:LD power

ratio; $P = 0.01$) (Fig. 1*B*, *E*; Table 1). This suggests that ageing differentially weakens circadian rhythms in wheel-running activity of female compared with male mice, with the LD cycle functioning to stabilise these rhythms.

To further investigate these sex- and age-related effects of the LD cycle on wheel running, the day–night distribution of rodent activity, as well as the time of peak activity and temporal precision of this peak, were examined. This revealed that middle-aged male and female mice engaged in more wheel running in the lights-on phase compared with the lights-off phase than young male and female mice ($P = 0.004$ and $P = 0.003$, respectively) (Fig. 2*A*–*C*; Table 1). In addition, the regularity in the time of the daily peak in wheel running (measured using vector length; Fig. 2*C*), did

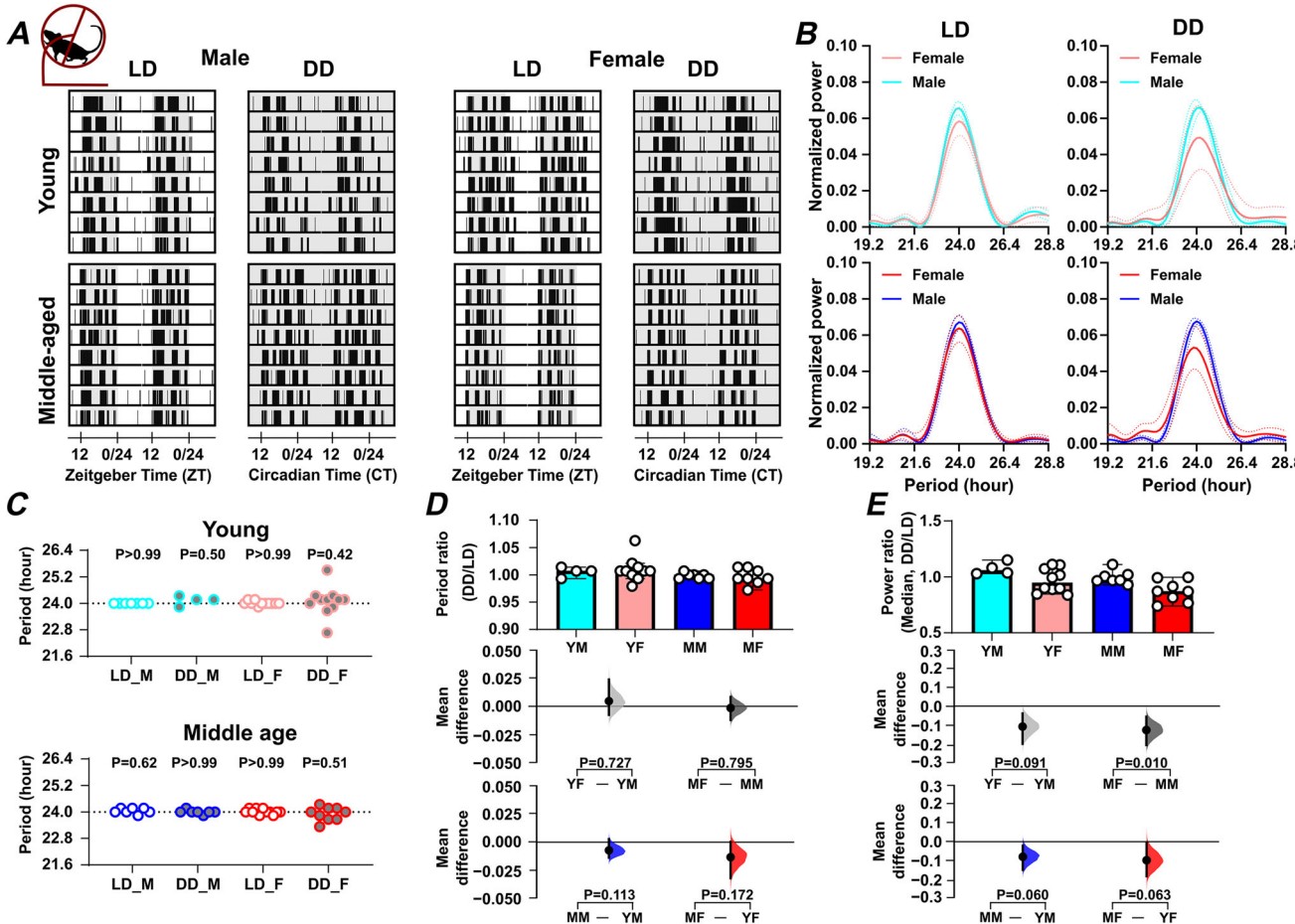

**Figure 1. Sex and age impact on daily and circadian rhythms in locomotor activity**
*A*, representative double-plotted actograms of wheel-running activity under light–dark (LD) and constant dark (DD) conditions, with shaded areas representing dark. *B*, Lomb-Scargle periodograms display the dominant circadian period and the strength of the rhythm in wheel-running activity. The continuous line represents the mean value, and the dotted lines indicate the standard deviation (SD). *C*, summary of daily/circadian period. The horizontal dotted line indicates the 24 h period. A Wilcoxon's signed-rank test was used to determine whether the circadian period deviated significantly from 24 h (*$P < 0.05$). Abbreviations: LD: light–dark cycle (standard 12 h light/12 h dark conditions); DD: constant dark (continuous dark conditions used to assess endogenous rhythms); M: male; F: female. Comparison of (*D*) the ratio of circadian/daily period (DD/LD) and (*E*) the ratio of circadian/daily rhythm power (DD/LD) across sex and age groups. Upper panel: Box plots showing medians with 95% confidence intervals. Middle panel: Comparisons focused on sex differences. Lower panel: Comparisons focused on age differences. The mean differences for comparisons are shown in Cumming estimation plots above each panel. Each mean difference is presented as a bootstrap sampling distribution, with mean differences depicted as dots and 95% confidence intervals indicated by the ends of the vertical error bars. Permutation *t* tests were used with 5000 bootstrap samples; confidence intervals are bias-corrected and accelerated. Reported *P* values represent the probability of observing the effect size (or greater), assuming the null hypothesis of zero difference. Group labels: YF = young female (*n* = 11); MF = middle-aged female (*n* = 9–11); YM = young male (*n* = 4–7); MM = middle-aged male (*n* = 7). For statistical details, please refer to the supplementary statistical summary file.

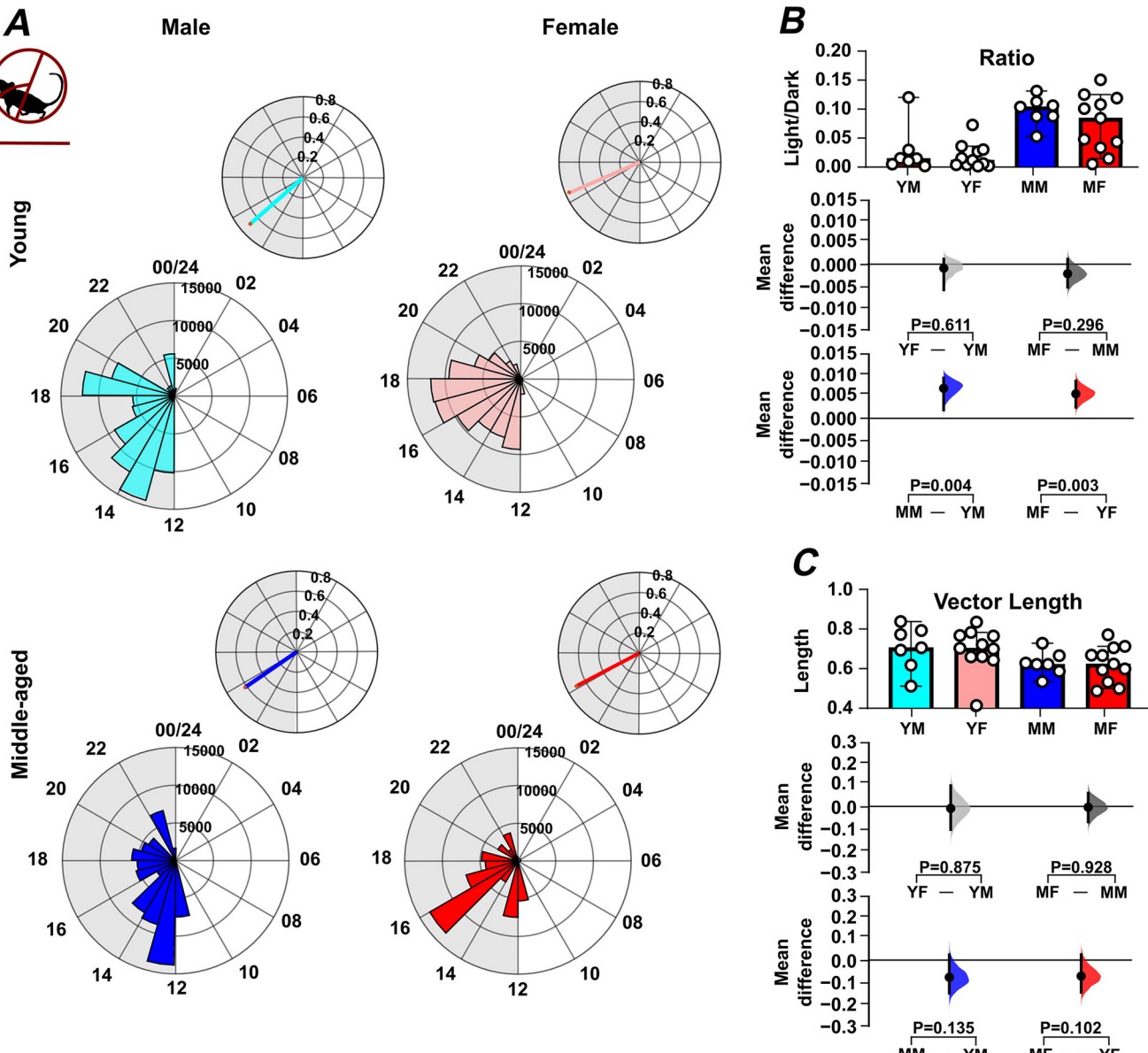

**Figure 2. Sex and age impact on the distribution of locomotor activity over the 24 h light–dark cycle**
*A*, representative daily activity patterns for wheel-running activity. Data points are aligned to the time of day and presented as a time-of-day (0–24 h) rose plot. This represents the mean activity over a 10 day recording epoch. Petals indicate time bins (time resolution in hours), with petal height reflecting the mean activity within each bin. The upper panel includes the mean resultant vector. Vector direction illustrating the peak timing of activity. The vector length indicates the strength of the rhythm. *B*, comparison of the ratio of activity during the light phase *versus* the dark phase across sex and age groups. *C*, comparison of vector length (indicating rhythmic strength) across sex and age groups. Upper panel: box plots showing medians with 95% confidence intervals. Middle panel: comparisons focused on sex differences. Lower panel: comparisons focused on age differences. Mean differences for comparisons are shown in Cumming estimation plots. Each mean difference is presented as a bootstrap sampling distribution, with mean differences depicted as dots and 95% confidence intervals indicated by the ends of the vertical error bars. Permutation *t* tests were used with 5000 bootstrap samples; confidence intervals are bias-corrected and accelerated. Reported *P* values represent the probability of observing the effect size (or greater), assuming the null hypothesis of zero difference. Group labels: YF = young female (*n* = 12); MF = middle-aged female (*n* = 13); YM = young male (*n* = 7); MM = middle-aged male (*n* = 8). For statistical details, please refer to the supplementary statistical summary file.

**Table 1. Summary of wheel-running results LD: light–dark cycle (12:12 h); DD: constant darkness. '⬈AGE' indicates a significant effect of age; '⚥' indicates a significant effect of sex. The data are presented as means ± SD.**

| | Weight change | Young male<br>No change | Young female<br>No change | Middle-aged male<br>Weight lost | Middle-aged female<br>Weight lost |
|---|---|---|---|---|---|
| Wheel running | Circadian period (LD): hour | 24.00 ± 0.00<br>(*n* = 7) | 24.00 ± 0.07<br>(*n* = 11) | 24.05 ± 0.13<br>(*n* = 7) | 23.98 ± 0.10<br>(*n* = 11) |
| | Circadian period (DD): hour | 24.12 ± 0.20<br>(*n* = 4) | 24.09 ± 0.66<br>(*n* = 11) | 24.00 ± 0.11<br>(*n* = 7) | 23.91 ± 0.3<br>(*n* = 9) |
| | Power of ratio (LD/DD) | 1.07 ± 0.06<br>(*n* = 4) | 0.96 ± 0.09<br>(*n* = 10) | 0.99 ± 0.05<br>(*n* = 7) | 0.87 ± 0.08<br>(*n* = 8)<br>⚥ |
| | Daily pattern | Well-entrained wheel-running rhythm.<br>(*n* = 7) | Well-entrained wheel-running rhythm.<br>(*n* = 12) | Daytime activity<br>(*n* = 8)<br>⬈AGE | Daytime activity<br>(*n* = 13)<br>⬈AGE |
| | Daily rhythm strength: vector length (between 0 and 1) | 0.69 ± 0.11<br>(*n* = 7) | 0.69 ± 0.10<br>(*n* = 12) | 0.62 ± 0.06<br>(*n* = 8) | 0.62 ± 0.11<br>(*n* = 13) |
| | Activity precision | Precise<br>(*n* = 7) | Variable<br>(*n* = 12)<br>⚥ ⬈AGE | Precise<br>(*n* = 8) | Precise<br>(*n* = 13) |
| | Duration of daily activity: minutes | 685.20 ± 41.06<br>(*n* = 7) | 678.00 ± 35.89<br>(*n* = 12) | 701.7 ± 44.80<br>(*n* = 8) | 746.4 ± 40.67<br>(*n* = 13)<br>⬈AGE |
| | 5 day rhythm: normalised power ($\times 10^{-4}$) | 1.63 ± 1.22<br>(*n* = 7) | 6.34 ± 3.67<br>(*n* = 12)<br>⚥ ⬈AGE | 3.58 ± 2.06<br>(*n* = 7) | 2.22 ± 3.22<br>(*n* = 11) |
| | 10 day rhythm: Normalised power ($\times 10^{-4}$) | 5.33 ± 3.15<br>(*n* = 7)<br>⚥ | 2.48 ± 1.73<br>(*n* = 12) | 5.45 ± 3.62<br>(*n* = 7)<br>⚥ | 2.33 ± 1.72<br>(*n* = 11) |

not significantly differ by age or sex, but there was a trend toward reduced regularity in this measure in middle-aged mice. This indicates that with increasing age, wheel-running activity in male and female mice is less confined to the lights-off phase.

Age and sex can also alter the capacity for and precision of daily voluntary exercise in rodents (Pastrick et al., 2024; Valentinuzzi et al., 1997) and analysis of these parameters indicated clear sex- and age-related differences in the duration as well as the timing of daily onset and offset of wheel-running activity (Fig. 3; Table 1). Middle-aged female mice exercised for longer each day than young female mice (*P* = 0.003) (Fig. 3*C*). At the same time, the onset of this daily wheel-running rhythm was more variable in young female mice compared with either young male mice or middle-aged female mice (*P* = 0.041 and *P* = 0.026, respectively) (Fig. 3*E*). Young female mice also showed more variability in the daily

offset of wheel running compared with young male mice (*P* = 0.02) (Fig. 3*G*). Thus, in addition to age-related changes in the distribution of daily wheel running, age and sex differentially influence the duration and precision of locomotor activity.

Mice housed under LD in the small-cage environment also exhibited a nocturnal pattern of daily activity (Fig. 3*A*) with many of the sex- and age-related differences noted above (duration of daily wheel running and variability in onset/offset) also observed (Fig. 3*B–F*, and *H*). One additional difference found was that middle-aged female mice engaged in wheel running for a longer duration per day than middle-aged male mice (*P* = 0.008) (Fig. 3*D*). Further, both middle-aged males

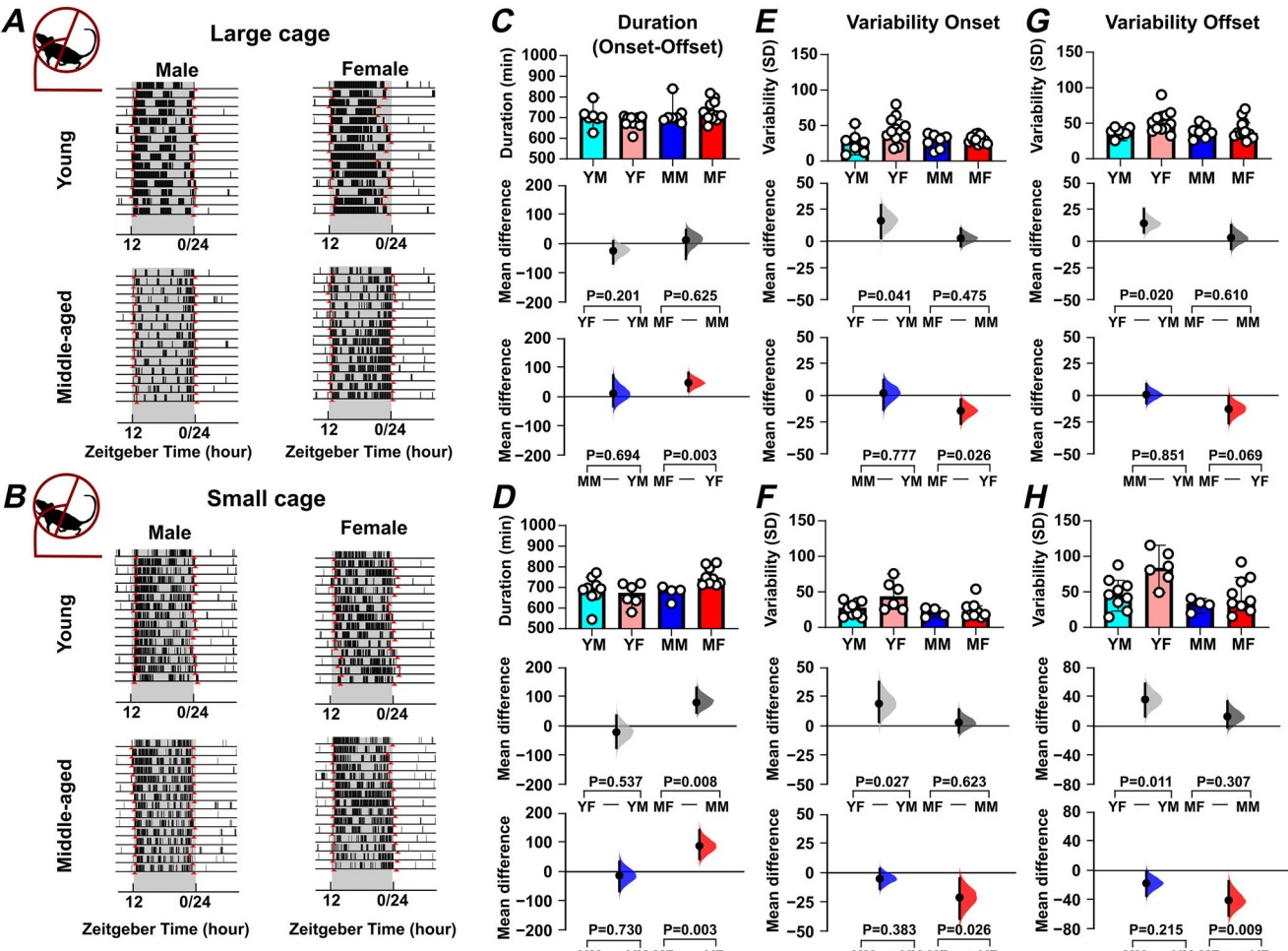

**Figure 3. Sex and age impact on the duration and precision of daily locomotor activity**
Duration of daily activity and its onset/offset vary with sex and age. *A*, *B*, example actogram recording obtained from mice housed in large cages and recorded using the TSE system (*A*) or housed in small cages and recorded with the ClockLab system (*B*), highlighting the onset and offset times (depicted in red) of locomotor activity. *C*, *D*, comparison of the duration of daily activity (from onset to offset times) across sex and age groups of mice measured with the TSE system (*C*) and ClockLab system (*D*). *E*, *F*, comparison of the variability in onset times, expressed as standard deviation (SD), across groups measured with the TSE system (*E*, large cage) and ClockLab system (*F*, small cage). *G*, *H*, comparison of the variability in offset times, expressed as SD, across mice recorded with the TSE system (*G*) or the ClockLab system (*H*). All comparisons were conducted using estimation statistics. Upper panel: box plots showing medians with 95% confidence intervals. Middle panel: comparisons focused on sex differences. Lower panel: comparisons focused on age differences. Mean differences are visualized in Cumming estimation plots positioned above each panel. Each mean difference is represented as a bootstrap sampling distribution (5000 bootstrap samples, bias-corrected and accelerated), with dots marking the mean differences and vertical error bars showing 95% confidence intervals. Permutation *t* tests were used, and reported *P* values indicate the probability of observing the effect size (or greater) under the null hypothesis of zero difference. Group labels: YF = young female (*n* = 12); MF = middle-aged female (*n* = 13); YM = young male (*n* = 7); MM = middle-aged male (*n* = 8). For statistical details, please refer to the supplementary statistical summary file.

**Table 2. Summary of drinking results** LD: light–dark cycle (12:12 h); DD: constant darkness. '' indicates a significant effect of age; '' indicates a significant effect of sex. The data are presented as means ± SD.

| | | Young male | Young female | Middle-aged male | Middle-aged female |
|---|---|---|---|---|---|
| Drinking | Circadian period (LD): hour | 23.95 ± 0.11 (*n* = 7) | 24.00 ± 0.0 (*n* = 11) | 23.95 ± 0.08 (*n* = 8) | 23.98 ± 0.13 (*n* = 11) |
| | Circadian period (DD): hour | 24.21 ± 0.28 (*n* = 4) | 24.19 ± 0.23 (*n* = 11) | 24.05 ± 0.11 (*n* = 8) | 24.09 ± 0.24 (*n* = 9) |
| | Power of ratio (LD/DD) | 1.04 ± 0.02 (*n* = 4)) | 1.01 ± 0.03 (*n* = 11) | 0.99 ± 0.03 (*n* = 7) | 0.97 ± 0.06 (*n* = 10) AGE |
| | Daily pattern | Well-entrained drinking rhythm (*n* = 7) | Well-entrained drinking rhythm (*n* = 12) | Well-entrained drinking rhythm (*n* = 8) | Daytime activity (*n* = 13) AGE |
| | Daily rhythm strength: vector length (between 0 and 1) | 0.49 ± 0.08 (*n* = 7) | 0.46 ± 0.14 (*n* = 12) | 0.48 ± 0.08 (*n* = 8) | 0.38 ± 0.11 (*n* = 13) AGE |
| | Activity precision | Precise (*n* = 7) | Precise (*n* = 12) | Precise (*n* = 8) | Precise (*n* = 13) |
| | Duration of daily activity: minutes | 686.8 ± 78.55 (*n* = 7) | 666.4 ± 101.33 (*n* = 12) | 742.5 ± 56.48 (*n* = 8) | 758.2 ± 80.15 (*n* = 13) AGE |
| | 5 day rhythm: normalised power ($*10^{-4}$) | 2.38 ± 2.53 (*n* = 7) | 2.92 ± 1083 (*n* = 12) | 2.75 ± 2.49 (*n* = 8) | 1.26 ± 1.48 (*n* = 13) AGE |
| | 10 day rhythm: normalised power ($*10^{-4}$) | 2.27 ± 2.14 (*n* = 7) | 2.69 ± 3.46 (*n* = 12) | 3.17 ± 1.50 (*n* = 8) | 1.66 ± 2.09 (*n* = 13) |

and females had significantly elevated activity during the lights-on phase of the 24 h LD cycle (Fig. S1*A*, *B*) with the regularity in the time of peak activity (vector length) also being reduced in middle-aged mice (*P* = 0.0046; Fig. S1*C*). This indicates that under LD, the housing conditions (cage and running wheel size) used in this study are not a key factor in shaping these wheel-running rhythms.

**Ingestive behaviour.** Parameters of ingestive behaviours, water and food intake, could only be assessed in mice housed in large cages. Under LD conditions, these mice showed the expected nocturnal pattern of drinking also observed for their wheel-running activity. However, sex-

and age-related differences were also observed in their water intake patterns (Figs 4 and 5; Table 2). While the period of drinking activity in young and middle-aged male and female mice did not differ from 24 h (Fig. 4*A*–*C*), middle-aged female mice had reduced regularity in the timing of peak drinking (vector length; *P* = 0.05) and engaged in drinking more during the lights-on phase than male mice of corresponding age (*P* = 0.004) (Fig. 5*A*–*D*). Unlike wheel running, the daily onset of drinking behaviour was more variable in middle-aged male mice compared with young male mice (*P* = 0.03), while similar to wheel running, the daily duration in drinking behaviour increased in middle-aged female mice

compared with young female mice ($P = 0.010$) (Fig. 6). Subsequent assessment of circadian drinking patterns in DD revealed that young female mice exhibited a free-running period in drinking activity that was longer than 24 h ($24.19 \pm 0.072$ h; $P = 0.01$), while the period of this rhythm did not differ much from 24 h in the other mouse groups (Fig. 4C). Relative to LD, the strength of

the drinking rhythm declined in DD more in middle-aged male mice than in young male mice ($P = 0.027$) (Fig. 4E). This indicates that the LD cycle acted to constrain the circadian period of drinking in young female mice and to bolster the rhythm in middle-aged male mice.

As with wheel running and drinking, mice exhibited a nocturnal pattern of food ingestion under LD (Figs 7A–C

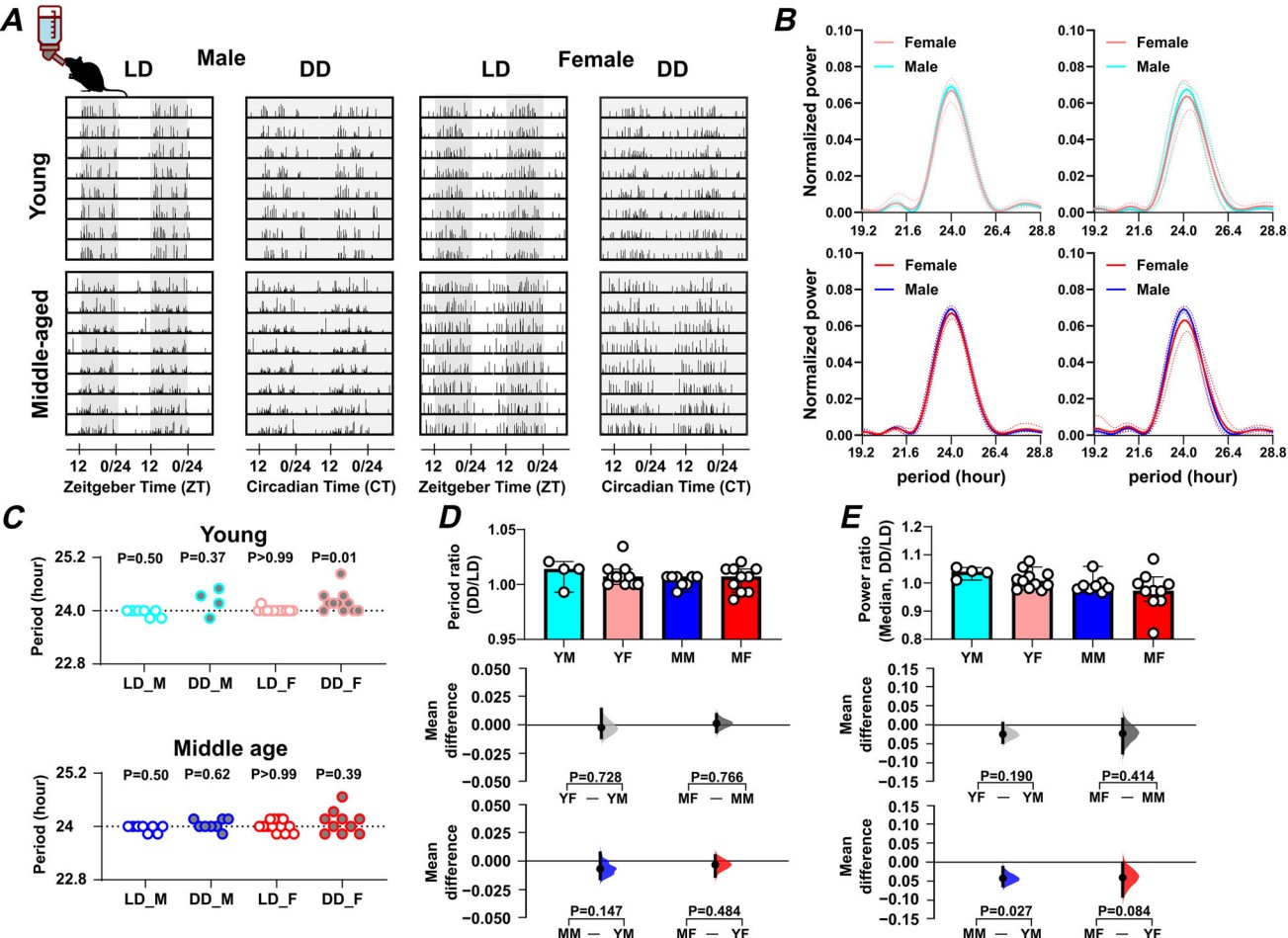

**Figure 4. Sex and age impact on daily rhythms in drinking activity**
*A*, representative double-plotted actograms of drinking activity under light–dark (LD) and constant darkness (DD) conditions, with shaded areas representing darkness. *B*, Lomb-Scargle periodograms display the dominant daily period and the strength of the rhythm in drinking activity. The continuous line represents the mean value, and the dotted lines indicate the standard deviation (SD). *C*, summary of circadian peak timing. The horizontal dotted line indicates the 24 h period. A Wilcoxon's signed-rank test was used to determine whether the circadian period deviated significantly from 24 h (*$P < 0.05$). Abbreviations: LD: light–dark cycle (standard 12 h light/12 h dark conditions); DD: constant dark (continuous dark conditions used to assess endogenous rhythms); M: male; F: female. Comparison of (*D*) the ratio of circadian/daily period (DD/LD) and (*E*) the ratio of circadian/daily rhythm power (DD/LD) across sex and age groups. Upper panel: box plots showing medians with 95% confidence intervals. Middle panel: comparisons focused on sex differences. Lower panel: comparisons focused on age differences. The mean differences for comparisons are shown in Cumming estimation plots above each panel. Each mean difference is presented as a bootstrap sampling distribution, with mean differences depicted as dots and 95% confidence intervals indicated by the ends of the vertical error bars. Permutation *t* tests were used with 5000 bootstrap samples; confidence intervals are bias-corrected and accelerated. Reported *P* values represent the probability of observing the effect size (or greater), assuming the null hypothesis of zero difference. Group labels: YF = young female (*n* = 11); MF = middle-aged female (*n* = 10–11); YM = young male (*n* = 4–7); MM = middle-aged male (*n* = 8). For statistical details, please refer to the supplementary statistical summary file.

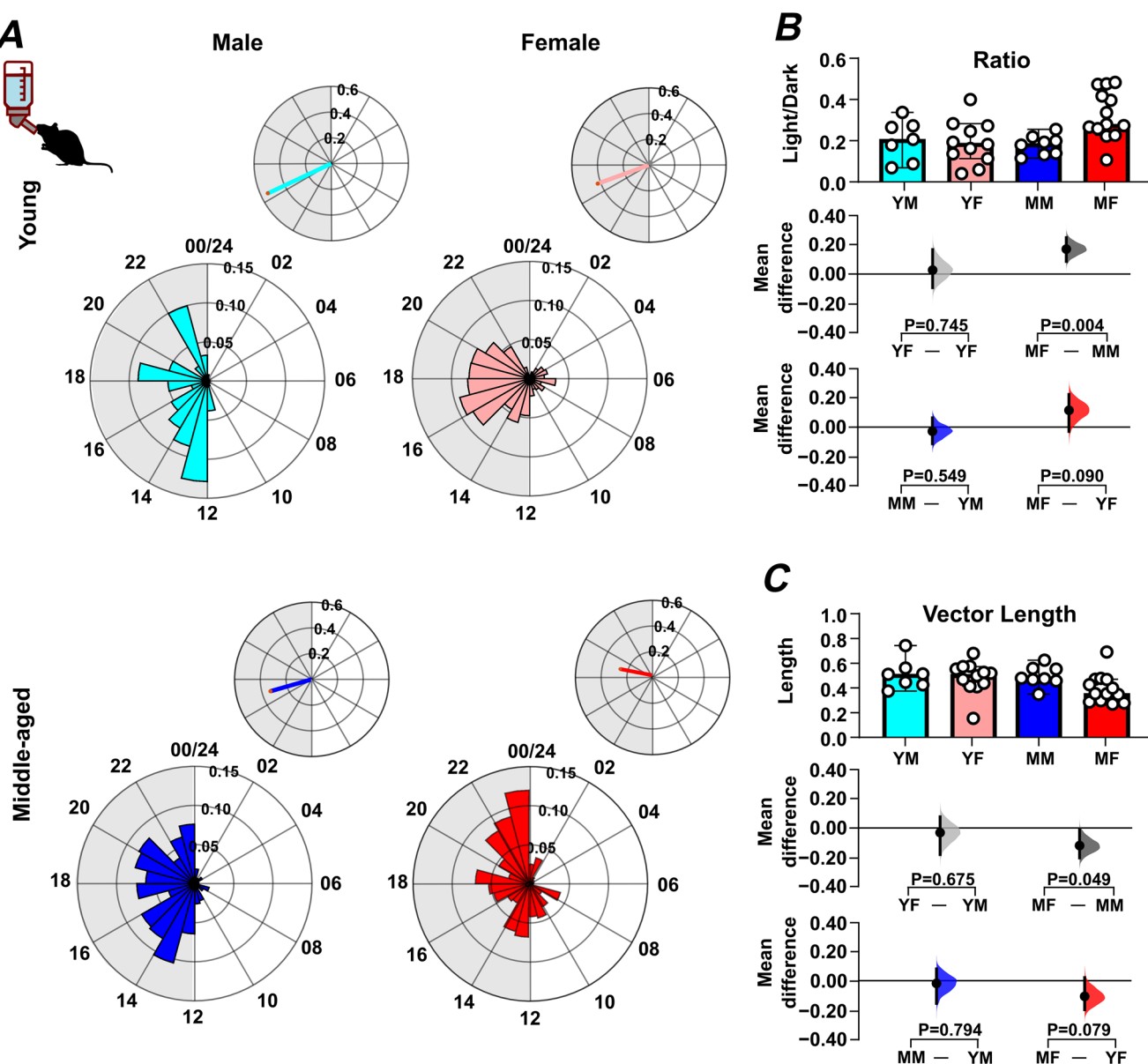

**Figure 5. Sex and age impact on daily activity patterns in drinking activity**
*A*, representative daily activity patterns for drinking activity. Data points are aligned to the time of day and presented as a time-of-day (0–24 h) rose plot. This represents the mean activity over a 10 day recording epoch. Petals indicate time bins (time resolution in hours), with petal height reflecting the mean activity within each bin. The upper panel includes the mean resultant vector. Vector direction illustrating the peak timing of activity. The vector length indicates the strength of the rhythm. *B*, comparison of the ratio of activity during the light phase *versus* the dark phase across sex and age groups. *C*, comparison of vector length (indicating rhythmic strength) across sex and age groups. Upper panel: box plots showing medians with 95% confidence intervals. Middle panel: comparisons focused on sex differences. Lower panel: comparisons focused on age differences. Mean differences for comparisons are shown in Cumming estimation plots. Each mean difference is presented as a bootstrap sampling distribution, with mean differences depicted as dots and 95% confidence intervals indicated by the ends of the vertical error bars. Permutation *t* tests were used with 5000 bootstrap samples; confidence intervals are bias-corrected and accelerated. Reported *P* values represent the probability of observing the effect size (or greater), assuming the null hypothesis of zero difference. Group labels: YF = young female (*n* = 12); MF = middle-aged female (*n* = 13); YM = young male (*n* = 7); MM = middle-aged male (*n* = 8). For statistical details, please refer to the supplementary statistical summary file.

and 8; Table 3), with the period of daily feeding behaviour not differing from 24 h in any sex/age combination of mice (Fig. 7*C*). Middle-aged male mice showed a significantly lower light–dark feeding ratio compared with both young male mice (*P* = 0.0002) and middle-aged female mice (*P* = 0.009), indicating that their food intake was more strongly concentrated during the dark (active) phase (Fig. 8*A* and *B*; Table 3). This heightened nocturnality in feeding behaviour may reflect a more rigid circadian regulation of metabolism in middle-aged male mice. They also had a greater regularity in the timing of peak food

ingestion daily rhythm in feeding behaviour than young male mice (vector length; *P* = 0.033) (Fig. 8*C*). As with wheel running and drinking behaviour, middle-aged female mice had a longer duration of daily feeding activity per 24 h than young female mice (*P* = 0.0006) (Fig. 6*F*; Table 3). To assess intrinsic circadian rhythmicity, the animals were transferred to DD and similar to drinking behaviour, young female mice exhibited a feeding rhythm with a period significantly longer than 24 h (24.22 ± 0.057 h; *P* = 0.007), while the period remained unchanged in all other combinations of sex and age

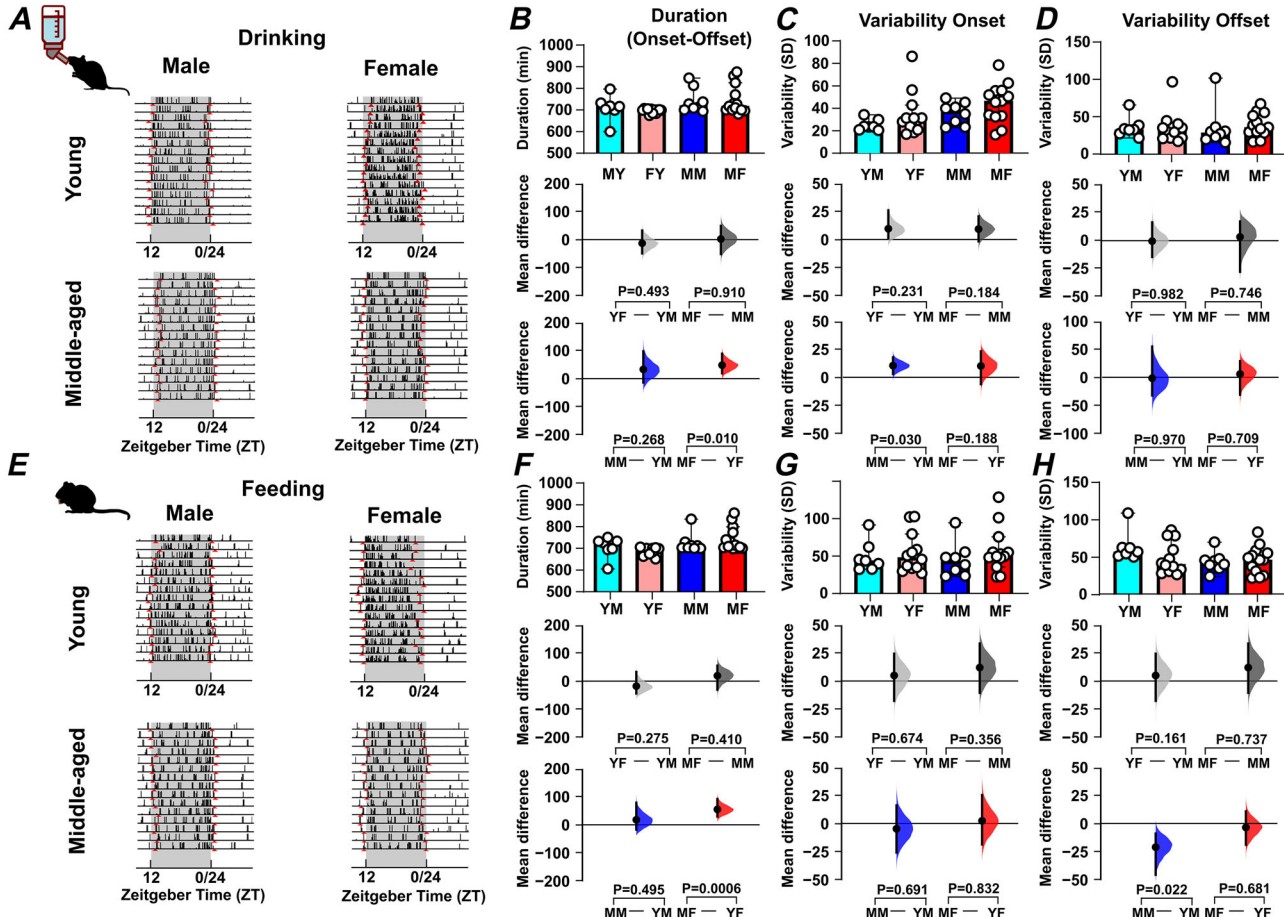

**Figure 6. Sex and age impact on the duration and precision of daily ingestive activity**
Differential sex- and age-related differences in daily drinking and feeding behaviours are seen in the duration as well as the variability (standard deviation, SD) in the onset and offset (depicted in red in the actograms). *A*, example actogram of drinking activity showing onset and offset times. *B–D*, comparison of drinking activity duration (onset to offset) (*B*), onset variability (*C*) and offset variability (*D*) across groups. *E*, example actogram of feeding activity showing onset and offset times. *F–H*, comparison of feeding activity duration (*F*), onset variability (*G*) and offset variability (*H*) across groups. All comparisons were conducted using estimation statistics. Upper panel: box plots showing medians with 95% confidence intervals. Middle panel: comparisons focused on sex differences. Lower panel: comparisons focused on age differences. Mean differences are visualized in Cumming estimation plots positioned above each panel. Each mean difference is represented as a bootstrap sampling distribution (5000 bootstrap samples, bias-corrected and accelerated), with dots marking the mean differences and vertical error bars showing 95% confidence intervals. Permutation *t* tests were used, and reported *P* values indicate the probability of observing the effect size (or greater) under the null hypothesis of zero difference group labels: YF = young female (*n* = 12); MF = middle-aged female (*n* = 13); YM = young male (*n* = 7); MM = middle-aged male (*n* = 8). For statistical details, please refer to the supplementary statistical summary file.

(Fig. 7*C*). Collectively, these analyses indicate that daily rhythms in ingestive behaviour are longer in duration in middle-aged female mice. In contrast, middle-aged male mice have increased variability in the onset and decreased variability in the offset compared with young male mice. Furthermore, in young female mice, the LD cycle constrains the period of both drinking and feeding rhythms to ∼24 h.

## Sex- and age-related differences in strength and prevalence of infradian rhythms in locomotor and ingestive activities

**Rhythms of 5 day periodicity.** The above analysis indicates age- and sex-related differences in daily rhythms, and to determine whether age and sex also influence infradian rhythms, we assessed rhythms associated with

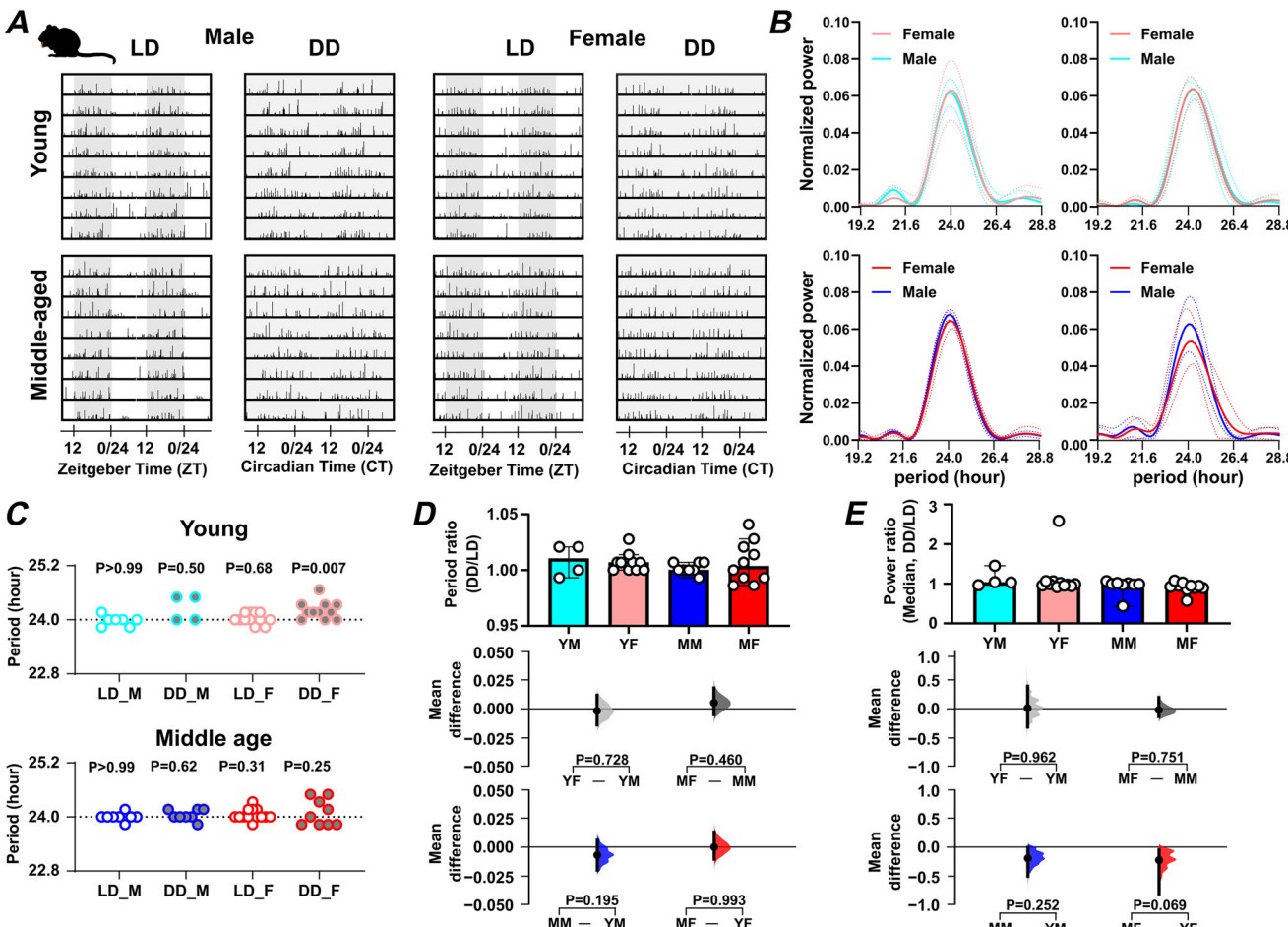

**Figure 7. Sex- and age-related impact on daily rhythms in feeding activity**
*A*, representative double-plotted actograms of feeding activity under light–dark (LD) and constant darkness (DD) conditions, with shaded areas representing dark. *B*, Lomb-Scargle periodograms displaying the dominant circadian period and rhythm strength of wheel-running activity. The thick line represents the mean value, and the dotted lines indicate the standard deviation (SD). *C*, summary of daily/circadian period. The horizontal dotted line indicates the 1 day (24 h) period. A Wilcoxon's signed-rank test was used to determine whether the circadian period deviated significantly from 24 h (\**P* < 0.05). Abbreviations: LD: light–dark cycle (standard 12 h light/12 h dark conditions), DD: constant dark (continuous dark conditions used to assess endogenous rhythms); M: Male; F: Female. Comparison of (*D*) the ratio of circadian/daily period (DD/LD) and (*E*) the ratio of circadian/daily rhythm power (DD/LD) across sex and age groups. Upper panel: box plots showing medians with 95% confidence intervals. Middle panel: comparisons focused on sex differences. Lower panel: comparisons focused on age differences. The mean differences for comparisons are shown in Cumming estimation plots above each panel. Each mean difference is presented as a bootstrap sampling distribution, with mean differences depicted as dots and 95% confidence intervals indicated by the ends of the vertical error bars. Permutation *t* tests were used with 5000 bootstrap samples; confidence intervals are bias-corrected and accelerated. Reported *P* values represent the probability of observing the effect size (or greater), assuming the null hypothesis of zero difference. Group labels: YF = young female (*n* = 11); MF = middle-aged female (*n* = 10); YM = young male (*n* = 4–7); MM = middle-aged male (*n* = 8). For statistical details, please refer to the supplementary statistical summary file.

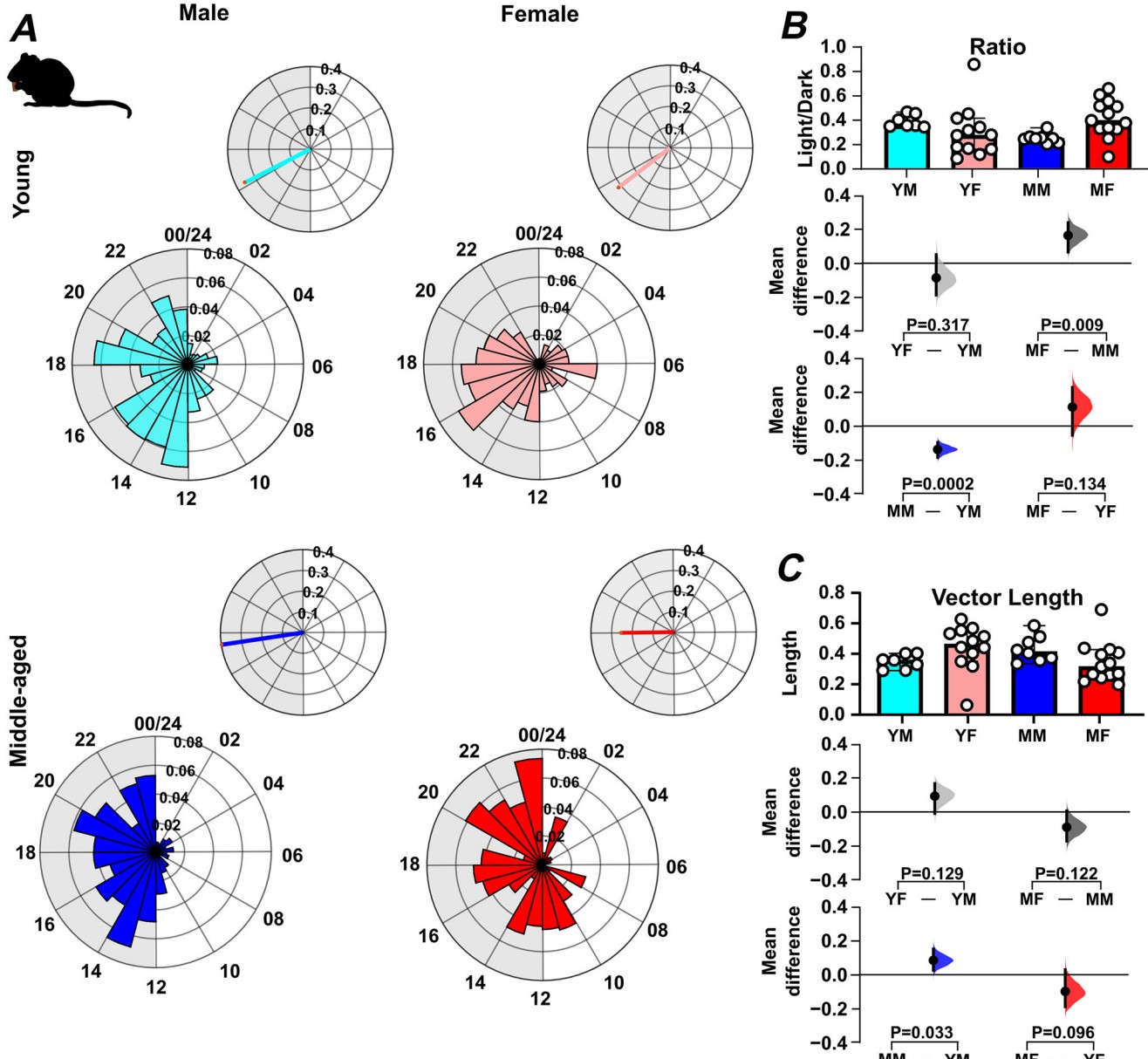

**Figure 8. Sex and age impact on daily activity patterns in feeding activity**

*A*, representative daily activity patterns for feeding activity. Data points are aligned to the time of day and presented as a time-of-day (0–24 h) rose plot. This represents the mean activity over a 10 day recording epoch. Petals indicate time bins (time resolution in hours), with petal height reflecting the mean activity within each bin. The upper panel includes the mean resultant vector. Vector direction illustrating the peak timing of activity. The vector length indicates the strength of the rhythm. *B*, comparison of the ratio of activity during the light phase *versus* the dark phase across sex and age groups. *C*, comparison of vector length (indicating rhythmic strength) across sex and age groups. Upper panel: box plots showing medians with 95% confidence intervals. Middle panel: comparisons focused on sex differences. Lower panel: comparisons focused on age differences. Mean differences for comparisons are shown in Cumming estimation plots. Each mean difference is presented as a bootstrap sampling distribution, with mean differences depicted as dots and 95% confidence intervals indicated by the ends of the vertical error bars. Permutation *t* tests were used with 5000 bootstrap samples; confidence intervals are bias-corrected and accelerated. Reported *P* values represent the probability of observing the effect size (or greater), assuming the null hypothesis of zero difference. Group labels: YF = young female (*n* = 12); MF = middle-aged female (*n* = 13); YM = young male (*n* = 7); MM = middle-aged male (*n* = 8). For statistical details, please refer to the supplementary statistical summary file.

**Table 3. Summary of feeding results.** LD: light–dark cycle (12:12 h); DD: constant darkness. '⬈AGE' indicates a significant effect of age; '⚥' indicates a significant effect of sex. The data are presented as means ± SD.

| | | Young male | Young female | Middle-aged male | Middle-aged female |
|---|---|---|---|---|---|
| Feeding | Circadian period (LD): hour | 24.05 ± 0.11 (*n* = 7) | 24.05 ± 0.10 (*n* = 11) | 24.00 ± 0.08 (*n* = 8) | 24.05 ± 0.13 (*n* = 10) |
| | Circadian period (DD): hour | 24.24 ± 0.28 (*n* = 4) | 24.22 ± 0.20 (*n* = 11) | 24.05 ± 0.11 (*n* = 8) | 24.22 ± 0.47 (*n* = 10) |
| | Power of ratio (LD/DD) | 1.12 ± 0.22 (*n* = 4) | 1.13 ± 0.50 (*n* = 11) | 0.93 ± 0.20 (*n* = 8) | 0.90 ± 0.13 (*n* = 10) |
| | Daily pattern | Active-phase dominant feeding (*n* = 7) | Active-phase dominant feeding (*n* = 12) | High phase specificity *N* = 8 ⚥ ⬈AGE | Active-phase dominant feeding (*n* = 13) |
| | Daily rhythm strength: vector length (between 0 and 1) | 0.34 ± 0.05 (*n* = 7) | 0.44 ± 0.14 (*n* = 12) | 0.43 ± 0.08 (*n* = 8) ⬈AGE | 0.34 ± 0.14 (*n* = 13) |
| | Activity precision | Precise (*n* = 7) | Precise (*n* = 12) | Precise (*n* = 8) | Precise (*n* = 13) |
| | Duration of daily activity: minutes | 689.9 ± 77.76 (*n* = 7) | 656.8 ± 108.05 (*n* = 12) | 724.4 ± 45.74 (*n* = 8) | 754.6 ± 69.30 (*n* = 13) ⬈AGE |
| | 5 day rhythm: Normalised power ($\times 10^{-4}$) | 2.96 ± 2.30 *n* = 7 | 6.61 ± 4.92 (*n* = 12) ⚥ ⬈AGE | 2.66 ± 1.27 (*n* = 7) | 2.14 ± 2.12 (*n* = 11) |
| | 10 day rhythm: Normalised power ($\times 10^{-4}$) | 3.84 ± 2.70 (*n* = 7) | 2.89 ± 1.94 (*n* = 12) | 5.77 ± 1.77 (*n* = 7) ⚥ ⬈AGE | 2.65 ± 2.92 (*n* = 11) |

the oestrous cycle (5 day periodicity) (Fig. S2) as well as cage-cleaning frequency (10 day periodicity). For animals housed in large cages, 5 day rhythms in wheel running were detected in 83% of young females (10 of 12 mice) but were infrequent in middle-aged females (1 of 11), young males (1 of 6) and middle-aged males (0 of 7) (Fig. S3*A*,*B*). Young female mice had stronger rhythms than young male mice (*P* = 0.002) and middle-aged female mice (*P* = 0.014) (Fig. 9*A* and *B*, Fig. S4 and S5; Table 1). In young female mice, but not their middle-aged counterparts, the strength of these 5 day rhythms was positively correlated with the variability in daily wheel-running onset and offset, indicating that this infradian rhythm is associated with alterations in the parameters of the daily/circadian rhythm (Fig. S6). For animals housed in small cages, 5 day rhythms in wheel running were detected in 83% of young females (5 of 6) but were infrequent in middle-aged females (2 of 9), young males (0 of 9) and middle-aged males (0 of 4) (Fig. S3*A*,*C*). Similar sex- and age-related differences to those seen in animals housed in large cages were observed in the strength of wheel-running rhythms: young female *vs*. young male mice (*P* = 0.001) and young female *vs*. middle-aged female mice (*P* = 0.006). While young male mice did not

exhibit consistent 5 day rhythms, middle-aged male mice showed a significant increase in 5 day rhythm strength ($P = 0.003$). This suggests that although the 5 day rhythm was absent or weak in young male mice, it emerged or became more prominent with age (Fig. 9$E$ and $F$). For 5 day rhythms in drinking and feeding (animals in large cages only), sex- and age-related differences were less frequent than for wheel running, with young female mice exhibiting stronger rhythms in drinking and feeding than middle-aged females ($P = 0.042$ and $P = 0.014$, respectively) (Fig. 10$A$, $B$, $E$, and $F$; Tables 2 and 3). Thus, young female mice had stronger 5 day infradian

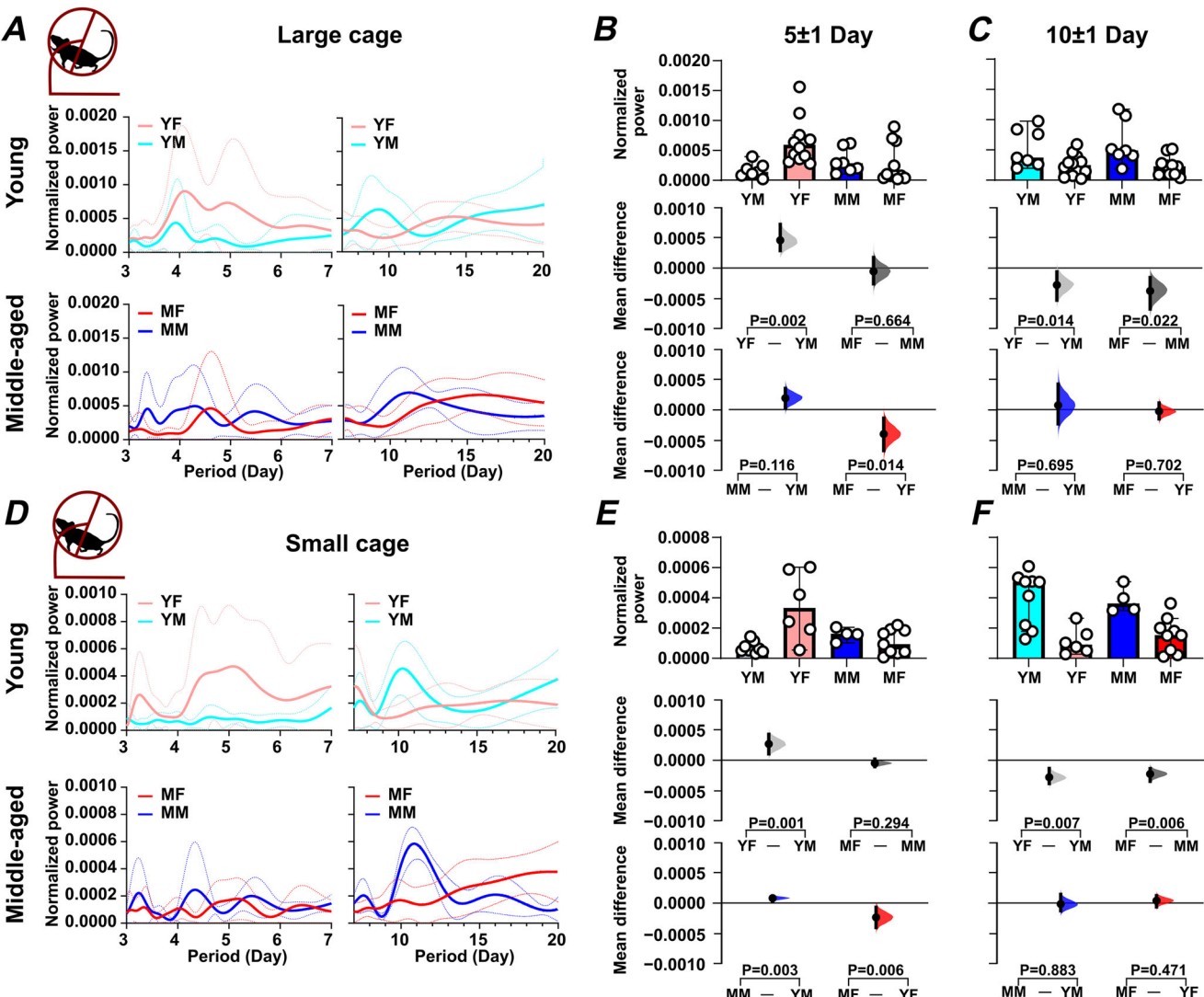

**Figure 9. Sex and age impact on infradian rhythms in locomotor activity**
Infradian rhythms (5 day and 10 day periodicity) in wheel-running activity using data from animals recorded in large and small cages. *A–D*, data from mice housed in large cages (TSE system.) *E–H*, data from mice housed in the small cages (ClockLab system). *A, D*, periodograms were generated using the Lomb-Scargle method. Solid lines indicate the mean power spectrum, and dotted lines represent the standard deviation. *B, E*, comparison of the power of 5 day period across groups. *C, F*, comparison of the power of 10 day period across groups. *B, C* and *E, F*, Upper panel: box plots showing medians with 95% confidence intervals. Middle panel: comparisons focused on sex differences. Lower panel: comparisons focused on age differences. All comparisons were performed using estimation statistics. Mean differences are visualized in Cumming estimation plots above each panel, represented as bootstrap sampling distributions (5000 samples, bias-corrected and accelerated). Dots indicate mean differences, with 95% confidence intervals shown as vertical error bars. Permutation $t$ tests were used, with $P$ values indicating the likelihood of observing the effect size (or greater) under the null hypothesis of zero difference. Group labels: YF = young female ($n = 12$); MF = middle-aged female ($n = 11$); YM = young male ($n = 7$); MM = middle-aged male ($n = 7$). For statistical details, please refer to the supplementary statistical summary file.

rhythms in wheel running and ingestive behaviour than middle-aged female mice and had stronger wheel-running rhythms than young male mice.

**Rhythms of 10 day periodicity.** For 10 day wheel-running rhythms, young and middle-aged male mice had stronger

rhythms than age-matched female mice ($P = 0.014$ and $P = 0.022$, respectively) (Fig. 9*A* and *C* and Fig. S5). More pronounced differences in wheel-running behaviour were observed in animals housed in small cages: young males *vs.* young females ($P = 0.007$) and middle-aged males *vs.* middle-aged females ($P = 0.006$) (Fig. 9*E* and *G*;

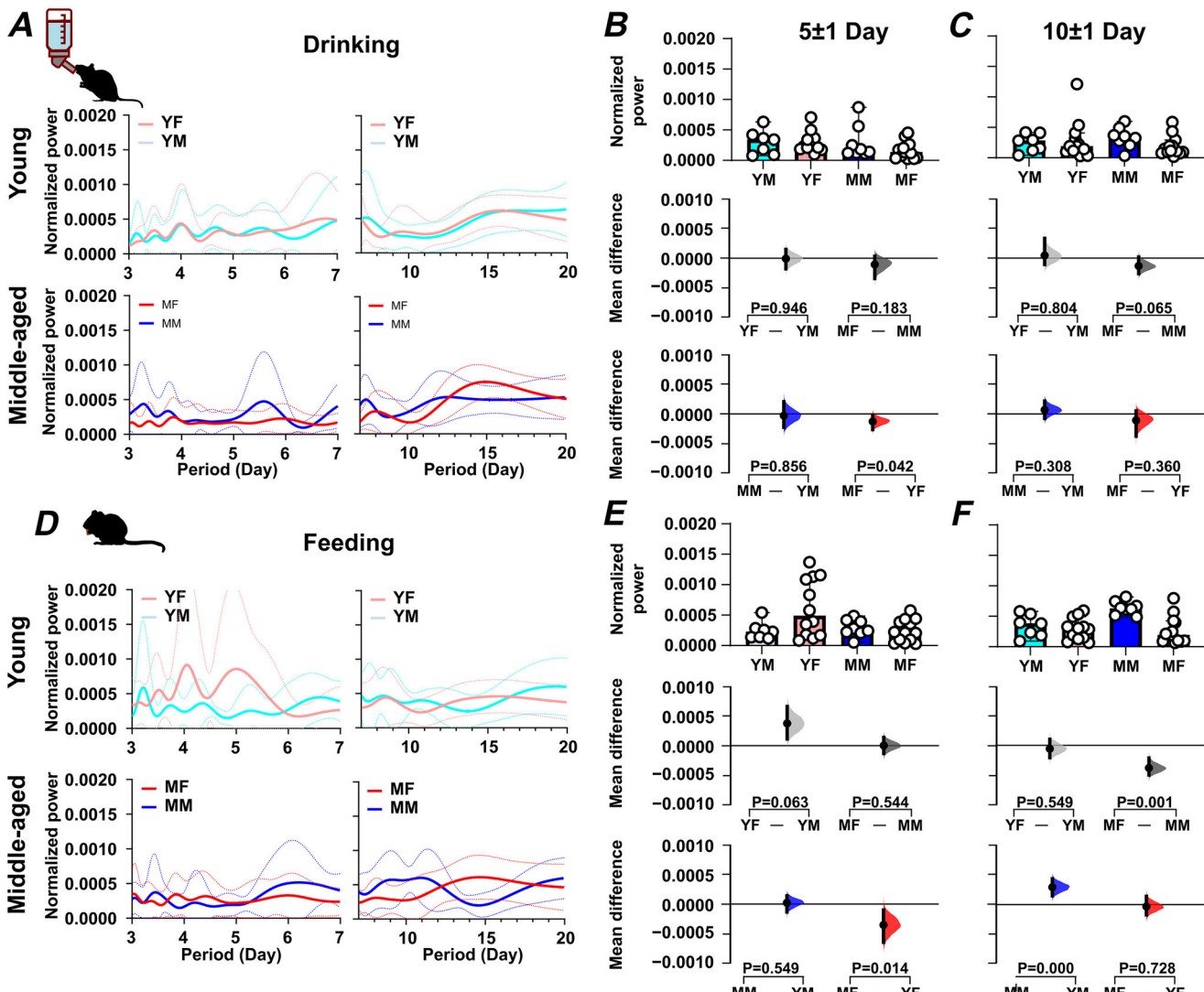

**Figure 10. Sex and age impact on infradian rhythms in ingestive activities**
Infradian rhythms (5 day and 10 day periodicity) in drinking and feeding activity vary with age and sex. *A–C* present data on drinking activity, while panels *D–F* focus on feeding activity. In (*A*) and (*D*), periodograms were generated using the Lomb-Scargle method. Solid lines indicate the mean power spectrum, and dotted lines represent the standard deviation. The power of the 5 day rhythm is compared across groups in (*B*) and (*E*), while (*C*) and (*F*) compare the power of the 10 day rhythm. The comparing period power (*B, C*) and (*E, F*) are divided into three sections. The upper panel features box plots displaying medians with 95% confidence intervals, while the middle panel focuses on comparisons of sex differences. The lower panel emphasizes comparisons of age differences. Statistical analyses were performed using estimation statistics, with mean differences visualized in Cumming estimation plots positioned above each panel. These plots represent bootstrap sampling distributions derived from 5000 samples (bias-corrected and accelerated). Dots indicate the mean differences, and vertical error bars display the 95% confidence intervals. Permutation *t* tests were applied, with *P* values showing the likelihood of observing the effect size (or greater) under the null hypothesis of zero difference. Group labels: YF = young female (*n* = 12); MF = middle-aged female (*n* = 13); YM = young male (*n* = 7); MM = middle-aged male (*n* = 8). For statistical details, please refer to the supplementary statistical summary file.

Table 1), indicating that male mice exhibited more robust 10 day wheel-running rhythms. We also examined the prevalence of significant 10 day rhythms across sex and age groups, focusing on animals housed in small cages. Among these, 55.5% (5 of 9) of young male mice and 16.7% (1 of 6) of young females exhibited significant 10 day rhythms, while 11.1% (1 of 9) of middle-aged females and 50% (2 of 4) of middle-aged males exhibited such rhythms. These proportions varied by sex (Fisher's exact test, $P = 0.042$) but not age (Fisher's exact test, $P = 0.435$, Fig. S7*A–C*). A similar trend in the 10 day rhythm was also observed in animals housed in the large cages; however, it did not reach the threshold for statistical significance ($P > 0.05$). These results suggest that the expression of 10 day rhythms is influenced not only by sex, but also by environmental factors such as cage size. For drinking, no sex- or age-related differences were found (Fig. 10*A* and *C*), but for feeding, middle-aged male mice had stronger rhythms than middle-aged female mice ($P = 0.0014$) and young male mice ($P = 0.0008$) (Fig. 10*E* and *H*; Tables 2 and 3). This indicates that middle-aged male mice exhibit stronger 10 day rhythms in wheel running and feeding than young male or middle-aged female mice.

The 10 day behavioural rhythms observed in male mice align with the timing of routine cage changes, which can disrupt normal behaviour in mice (Moore et al., 2024). To explore whether cage cleaning contributes to the observed 10 day rhythm, we examined the effects of altering cage-change frequency from every 10 days to every 7 days on wheel-running rhythms of male mice (housed in small cages). We found no significant differences in the strength of 1, 7 or 10 day wheel-running rhythms between the two cage-change schedules (Fig. S8*A–D*). However, none of the mice (0 out of 5) displayed significant 10 day rhythms under the 7 day cage-change condition (Fig. S7*D–F*). As cage change arouses mice, we further explored the impact of this by excluding the first 3 h of wheel running recorded after each cage change in the LS analysis, which tolerates short gaps. The ~10 day infradian rhythm remained evident even after this adjustment, indicating that the rhythm cannot be attributable to arousal-elicited wheel running (Fig. S8*E–G*). Therefore, these findings suggest that cage-change frequency alone is not sufficient to drive or wholly determine the strength of 7–10 day locomotor rhythms in male mice, but that the arousal/disruption associated with cage change is a contributor to the strength of infradian rhythms.

## Correlation between circadian and infradian rhythms across feeding, drinking and locomotor activities

The preceding analysis raises the possibility that relationships among the different periodicities in behavioural rhythms vary by sex and age. To evaluate this possibility, we analysed correlations among daily and infradian rhythms (5 day and 10 day periodicity) across feeding, drinking and locomotor activities (data from animals in large cages only). This revealed distinct patterns based on sex and age (Fig. 11*A–D*). In young male mice, no significant correlations (Fig. 11*A*) were observed. In contrast, young female mice exhibited substantial positive correlations between daily cycles in drinking and feeding, as well as between 10 day cycles in drinking and feeding activities (Fig. 10*B*). These findings suggest limited but consistent interactions between behavioural rhythms in young females.

In middle-aged animals, the correlation patterns became more complex, particularly in female mice. Male mice showed a positive correlation between 5 day cycles in drinking and feeding (Fig. 11*C*), whereas female mice showed a broader and more intricate network of correlations. This included positive correlations in daily rhythms in drinking and feeding, 5 day cycles in drinking and feeding, 5 day cycles in drinking and feeding together with 10 day cycles in wheel running (Fig. 11*D*). Negative correlations were also observed, including between daily rhythms in drinking and feeding with 10 day cycles in feeding (Fig. 11*D*). These findings highlight an age-related shift toward more interconnected and, in some cases, antagonistic relationships in female mice, suggesting sex-specific changes in rhythm coordination with age.

## Age and sex affect the ability to maintain body weight following transfer to single-housing conditions

The transfer of mice to single-housing conditions can result in weight loss (Muta et al., 2023). As our analysis indicates age- and sex-related differences in the rhythms of locomotor and ingestive behaviour, we next examined the animals' ability to maintain body weight following a change from group housing to single-housing conditions. Since wheel-running rhythms were similar primarily between animals housed in large and small cages and because preliminary analysis indicated that independent of cage size, mice in each of the sex/age groups lost similar amounts of body weight over the first 10 days in single housing (no statistically significant differences detected by *t* tests), we combined the body-weight data from both groups. Clear age-related differences were seen with young male and female mice generally maintaining their body weight over 70 days (Fig. 12*A* and *B*). In contrast, middle-aged male and female mice initially lost significant body weight 10–20 days after transfer to single-housing and did not recover to their initial body weight for another 50–60 days (Fig. 12*C* and *D*). The reduction in body weight was most prominent in

middle-aged female mice (Fig. 12*D*). Indeed, comparing the relationship between this change in body weight over the initial 10 days with the age of the mice indicated significant negative correlations for male ($P = 0.03$) and female ($P < 0.0001$) mice; the older the mouse the greater the loss in body weight in the first 10 days following transfer to single-housing conditions (Fig. 12*E* and *F*). This correlation of weight loss and age also differed between male and female mice ($P = 0.002$) (Fig. 12*G*). To gain insight into whether this loss in body weight is the result of a reduction in ingestion over the 10 day period, we compared food and water intake over this time period and other than weight loss in middle-aged females being larger than young males ($P = 0.0029$), we found no systematic age- or sex-related differences in these parameters (Fig. S9). These findings indicate a prominent sex difference in how the mice defend body weight following transfer to single-housing conditions and raise the possibility

that ageing amplifies physiological responses arising from exposure to an altered home-cage environment, without overtly altering the amount of food and water consumed.

## Correlations between daily and infradian rhythms across feeding, drinking and locomotor activities alter with age and sex

To further probe the relationship between body-weight loss in the first 10 days of single housing and behavioural rhythms, we analysed the correlations between age, sex, weight change, and the normalised power of 1 and 5 day periodicities in locomotor and ingestive activities (Summary provided in Table 4). For female mice, the normalised power of the daily pattern in wheel running (but not drinking or feeding behaviour; Fig. 13*A*, *C*, and *E*) increased with age and was positively correlated with

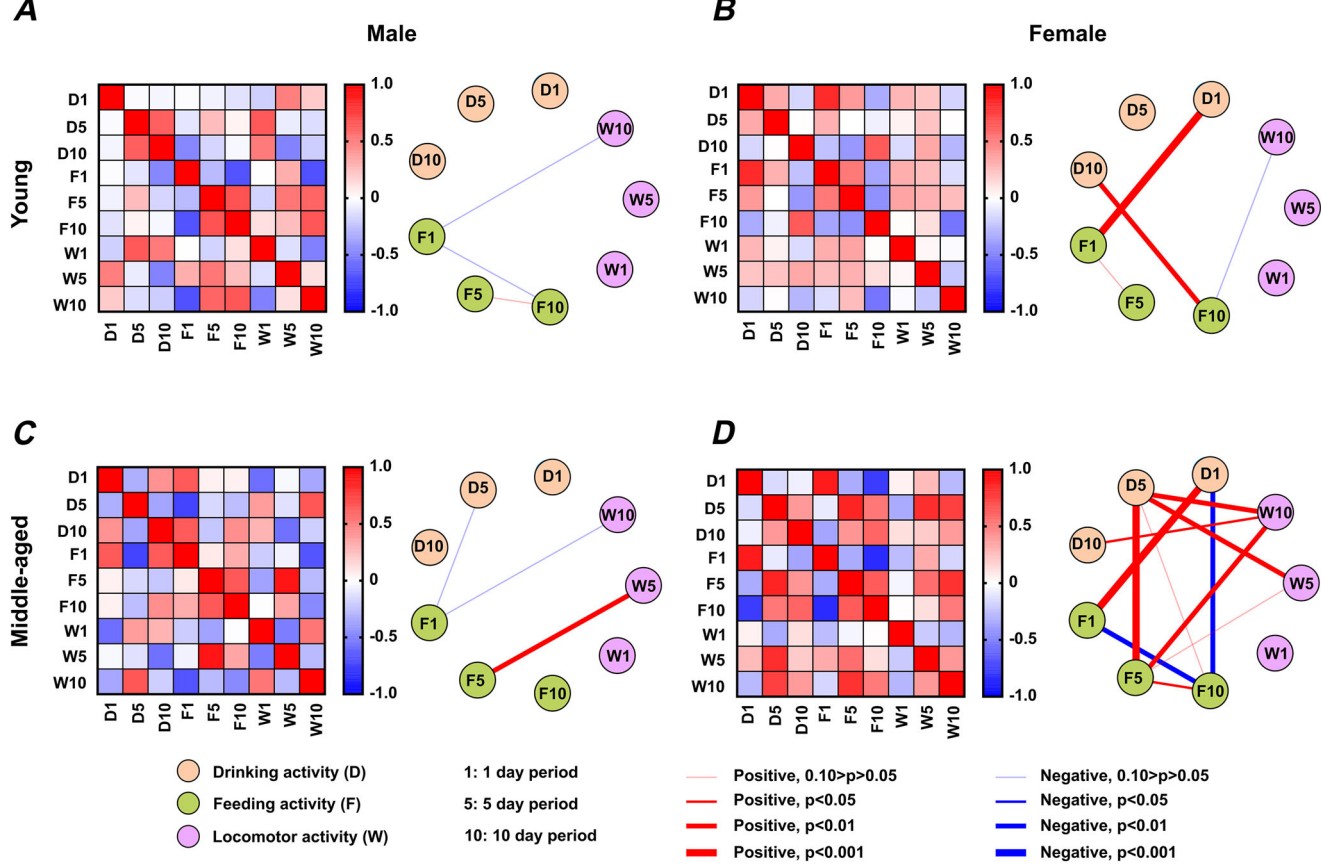

**Figure 11. Correlation between daily and infradian rhythms across behaviours**
Relationships between daily and infradian rhythms (5 day and 10 day periodicities) in feeding, drinking and locomotor activities increase in complexity with age in female mice. The heatmap on the left displays the correlation matrix for young males (*A*), young females (*B*), middle-aged males (*C*) and middle-aged females (*D*). Red indicates positive correlations, while blue represents negative correlations, with colour intensity reflecting the strength of the correlation. To the right of the heatmaps, a summary highlights statistically significant correlations, emphasizing key relationships within each group. This visualization highlights the interconnected nature of daily and infradian rhythms across behavioural domains. Young female (*n* = 12); middle-aged female (*n* = 11); young male (*n* = 7); middle-aged male (*n* = 7). For statistical details, please refer to the supplementary statistical summary file.

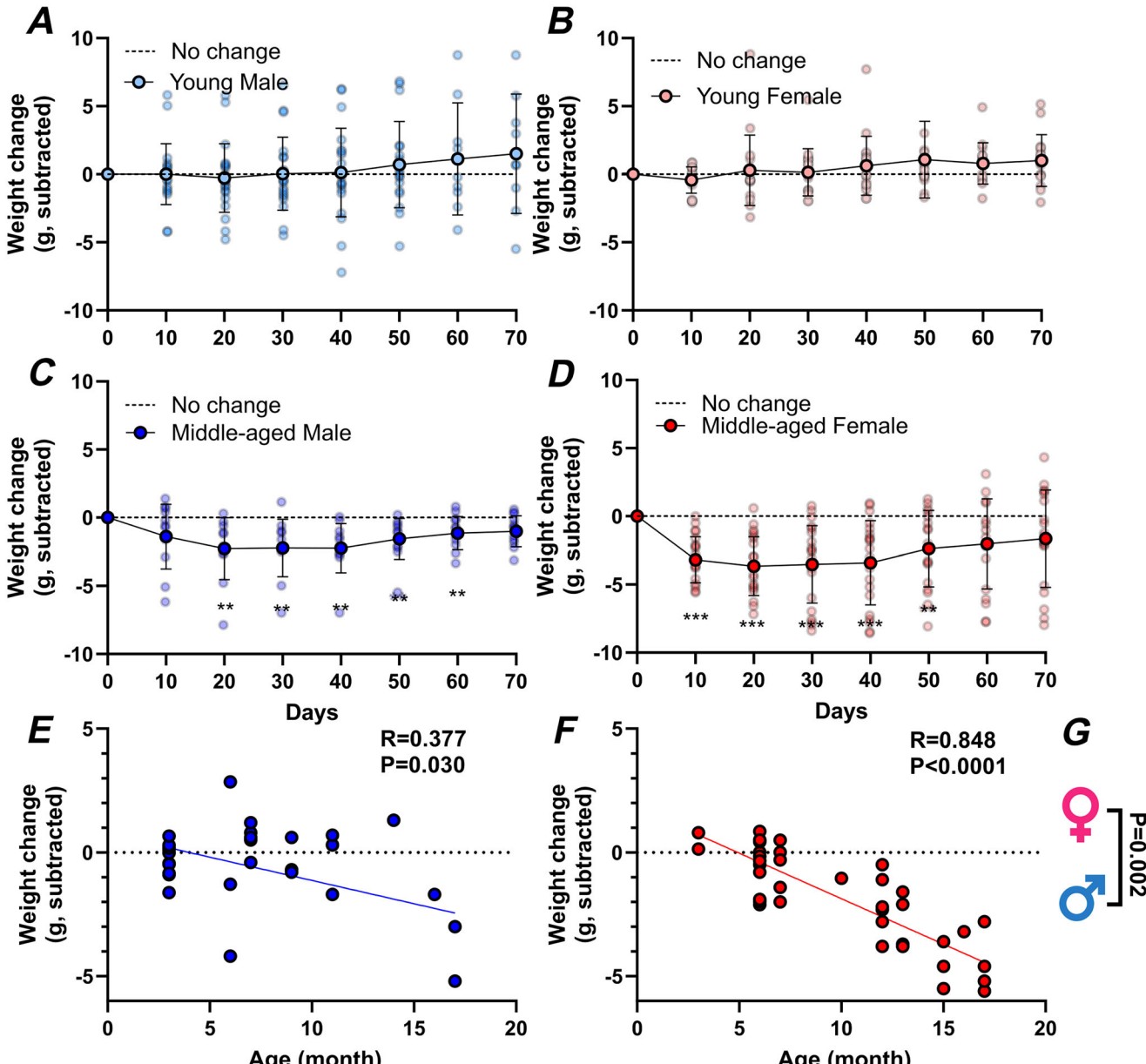

**Figure 12. Effects of single housing on body-weight maintenance**

*A–D*, the progression of body-weight change over 70 days in single housing of male and female animals across young and middle-aged groups. Animals were housed individually to ensure accurate monitoring. Body-weight change was calculated as the difference between each measurement day and the baseline weight recorded on Day 0. Data are presented as means ± SD. Group comparisons were performed using multiple Wilcoxon's tests with a 1% false discovery rate; **$P < 0.01$, *$P < 0.001$. *E* and *F*, Pearson's correlation analysis between age and weight change in males and females, respectively, highlighting age-related trends in body-weight dynamics. Pearson's correlation coefficients (*R*) were used to quantify the strength and direction of the relationships, with significant correlations denoted by corresponding *P* values. *G*, group comparisons of correlation coefficients were performed by transforming *R* values into *z*-scores using Fisher's *R*-to-*z* transformation, allowing for standardized comparisons across groups and conditions. Female (*n* = 38); Male (*n* = 33). For statistical details, please refer to the supplementary statistical summary file.

**Table 4. Correlations between age, weight change, and rhythm strength in daily and 5 day activity patterns. Values represent Pearson's correlation coefficients (*R*). (Male = 13; female = 23).**

|  |  | Male | Female |
|---|---|---|---|
| Correlation age *vs*. | Daily wheel running | 0.07 | 0.63 |
|  | Daily drinking | 0.47 | −0.43 |
|  | Daily feeding | 0.05 | 0.18 |
|  | 5 day wheel-running rhythm | 0.47 | −0.43 |
|  | 5 day drinking-running rhythm | −0.04 | −0.44 |
|  | 5 day feeding-running rhythm | 0.18 | −0.43 |
| Correlation weight change *vs*. | Daily wheel running | 0.62 | −0.62 |
|  | Daily drinking | −0.23 | −0.07 |
|  | Daily feeding | 0.04 | −0.05 |
|  | 5 day wheel-running rhythm | 0.10 | 0.44 |
|  | 5 day drinking-running rhythm | 0.30 | 0.46 |
|  | 5 day feeding-running rhythm | 0.04 | 0.46 |

age ($P = 0.0001$), whereas for the 5 day periodicity, the normalised power of the rhythm decreased with age and was negatively correlated with age ($P = 0.033$) (Fig. 13*B*, *D*, and *F*). No such correlations of daily or 5 day periodicity with age were found in male mice. This indicates differential effects of ageing on daily and 5 day rhythms in female mice and demonstrates that wheel-running activity is more sensitive to these effects than ingestive (both drinking and feeding) behaviours.

For changes in body weight in female mice, the normalised power of the daily rhythm in wheel running (but not feeding or drinking; Fig. 13*G*, *I*, and *K*) was negatively correlated with body-weight change ($R = -0.261$, $P = 0.002$), such that the stronger the daily rhythm, the larger the weight loss. An opposite correlation was seen with male mice, whereby the stronger the normalised power of the daily rhythm in wheel running (but not drinking or feeding; Fig. 13*I* and *K*), the smaller the weight loss ($P = 0.024$) (Fig. 13*G*). These correlations differed between male and female mice ($P = 0.0002$). This is consistent with the observation that food and water ingestion over the initial 10 days in single housing did not account for the change of body weight over this time period. In 5 day periodicity in female mice, the normalised power in drinking ($P = 0.029$), feeding ($P = 0.029$) and wheel-running ($P = 0.038$) activities was negatively correlated with a reduction in body weight (Fig. 13*H*, *J*, and *L*). This indicates that female mice expressing a strong 5 day rhythm will lose less body weight than those with weak 5 day rhythms in behaviour. No such relationships were found for male mice.

## Discussion

Here we reveal distinct age- and sex-dependent influences on daily and infradian rhythms of wheel running and ingestive behaviours as well as the ability of mice to maintain body weight under single-housing conditions. Behavioural rhythm complexity increased with age, particularly in females. In females, daily wheel-running rhythms strengthened with age, while 5 day rhythms across behaviours weakened; no such age effects were observed in males. Body-weight change following single housing was negatively associated with age in both sexes, being stronger in females. Rhythm–weight associations also differed by sex: in females, stronger daily rhythms predicted greater weight loss, whereas in males they predicted more minor loss; stronger 5 day rhythms in females were linked to less weight loss, with no equivalent effect in males. Additionally, 10 day rhythms emerged in small-cage conditions, persisting even when cage-change frequency was reduced from 10 to 7 days.

### Sex- and age-dependent variations in behavioural rhythms

A clear age effect was observed in daily wheel-running rhythms, with more variable effects on ingestive behaviours, particularly in females. Both middle-aged males and females showed increased daytime wheel running, but the rise in ingestive activity during the light phase was specific to middle-aged females. Middle-aged females also exhibited prolonged daily activity in both wheel running and ingestive behaviours. In contrast, young females displayed more variable onset and offset of these behaviours. In constant darkness, young females showed elongated circadian periods ($>24$ h) in feeding and drinking, but not in wheel running. By comparison, circadian periods in older females and in males remained close to 24 h, consistent with reports that homozygous PER2::LUC mice generally have slightly longer circadian periods ($\sim$24 h) than wild type C57BL6 animals ($\sim$23.6 h; Ralph et al., 2021).

These findings suggest age-related changes in circadian-regulated behaviours, potentially linked to internal timekeeping or metabolic demands. The discrepancy between locomotor and ingestive periods may reflect wheel running's engagement of reward pathways (Greenwood et al., 2011) not recruited during *ad libitum* feeding. Fewer age effects were seen in males: middle-aged males increased wheel running during lights-on but fed more during lights-off compared with young males. Overall, age-related rhythm changes were more pronounced in females.

The greater variability in young female rhythms likely reflects sensitivity to hormonal fluctuations and reliance on external light–dark cues to stabilise behaviour. This interpretation is supported by evidence from human and rodent studies (Ecochard, Stanford et al., 2024; Kuljis et al.,

2013; Mizuta et al., 2018; Vidafar et al., 2024). Moreover, sex differences in the SCN, including connectivity, neuropeptide signalling and hormone receptor expression (Abizaid et al., 2004; Bailey & Silver, 2014; Joye & Evans, 2022; Yan & Silver, 2016), may underlie the heightened variability in female mice. Together, these results indicate that the precision of daily rhythms in females is more dependent on external cues than in males.

### Sex- and age-related differences in the strength and prevalence of infradian activity rhythms

Young female mice exhibited significantly stronger 5 day rhythms in wheel-running activity, consistent with previous studies showing that locomotor rhythms in female

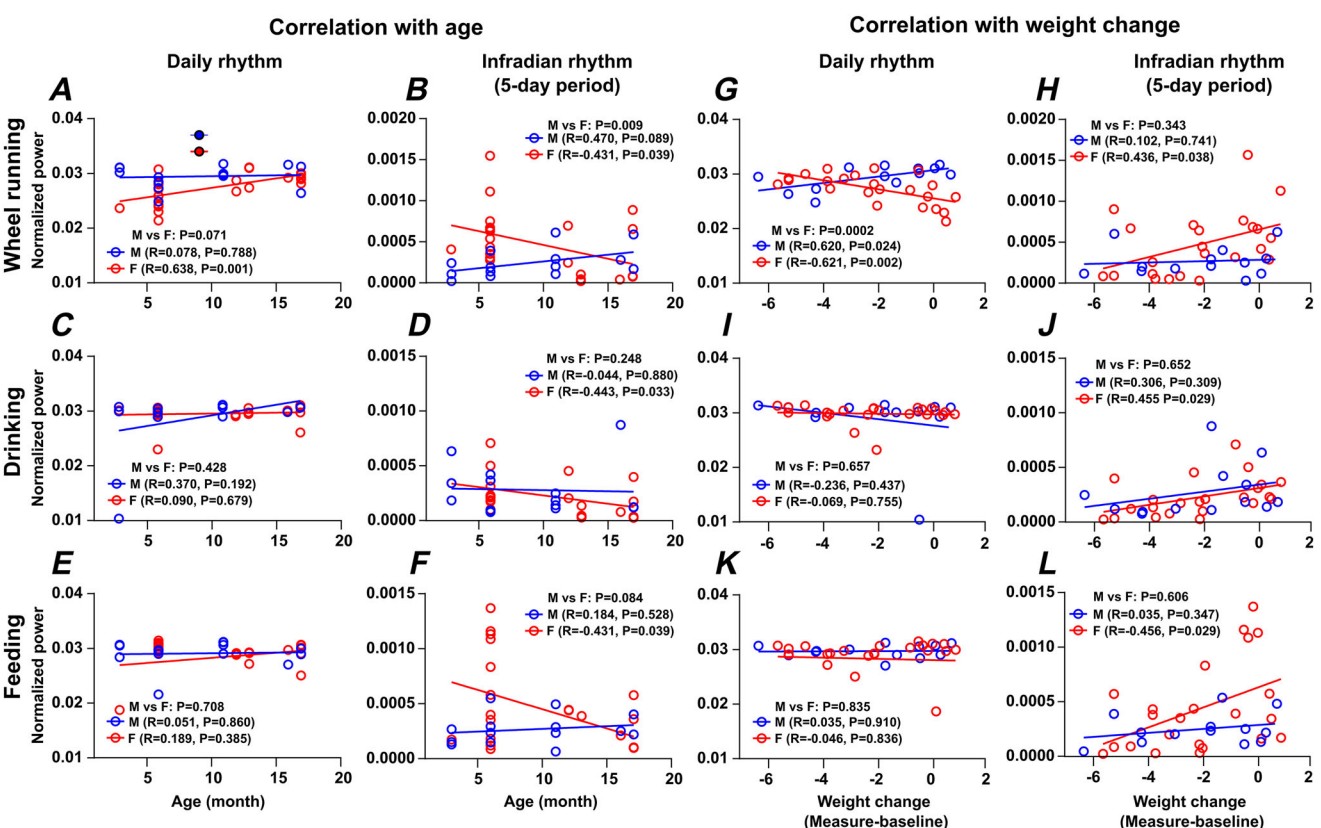

**Figure 13. Correlation between age, body-weight change and daily/infradian rhythms in locomotor, drinking and feeding activities**
Pearson's correlation analysis of age (*A–F*) and body-weight change (*G–L*) with daily and infradian (5 day period) rhythms across wheel-running, drinking and feeding activities reveals distinct sex and age differences. *A*, *C* and *E* show the correlation between age and daily rhythm strength in locomotor, drinking and feeding activity, respectively. *B*, *D* and *F* display the correlation between age and 5 day period strength in these activities in locomotor, drinking and feeding activity, respectively. Age was analysed as a continuous variable (in months), to capture within-group variability. *G*, *I* and *K* show the correlation between weight change and daily rhythm strength in locomotor, drinking and feeding activity, respectively. *H*, *J* and *L* show the correlation between weight change and 5 day period strength in locomotor, drinking and feeding activity, respectively. Pearson's correlation coefficients (*R*) indicate the strength and direction of the relationships, with significant correlations marked by corresponding *P* values. *R* values were transformed into *z*-scores using Fisher's *R*-to-*z* transformation to compare the correlation coefficients. This approach enables standardized comparisons across groups and conditions. Female (*n* = 23); Male (*n* = 13). For statistical details, please refer to the supplementary statistical summary file.

rodents cycle with a 4–5 day periodicity linked to the oestrous cycle (Alvord & Pendergast, 2024). Fluctuations in oestrogen levels regulate daily and circadian rhythms of activity, with activity onset advancing during pro-oestrus and oestrus when oestrogen peaks (Albers et al., 1981; Morin et al., 1977; Takasu et al., 2015). In our study, 5 day fluctuations in the onset and offset of wheel running were clear in young but diminished in middle-aged females, consistent with the reported reproductive senescence of laboratory mice at around 12 months of age (Koebele & Bimonte-Nelson, 2016). Variability in activity timing was strongly associated with 5 day rhythm strength in young but not middle-aged females. Vaginal cytology confirmed that middle-aged females lacking 5 day behavioural rhythms also lacked typical oestrus-associated cytological changes, further supporting reduced reproductive capability. Together, these findings indicate that robust 5 day rhythms in young females are tied to the oestrous cycle and largely absent in middle-aged animals.

Unexpectedly, we found a 10 day periodicity in locomotor activity, most pronounced in males housed in small cages. This coincided with the 10 day cage-change cycle, a known behavioural stimulus (Ratuski et al., 2024). When the schedule was shifted to 7 days, the number of mice showing a 10 day rhythm was abolished, though rhythm strength was unchanged. Removing them for the first 0–3 h after cage cleaning also did not alter the rhythm, indicating that external cues contribute but do not fully explain it. These findings point to additional regulatory factors, possibly an intrinsic ~10 day oscillator or longer-term homeostatic processes. The male predominance suggests sex-specific influences, potentially hormonal, metabolic, or behavioural. Such infradian cycles may shape baseline activity and should be considered when designing and interpreting longitudinal studies.

Correlations between daily and infradian rhythms in feeding, drinking and locomotor activity showed clear age- and sex-dependent patterns. Young males displayed no significant associations, whereas young females showed strong links between eating and drinking, suggesting a tighter coupling early in life. With ageing, correlations emerged in males, while middle-aged females showed even more complex interdependencies, indicating progressive integration of behavioural rhythms with age. These changes may reflect sex-specific influences of hormones and sex chromosomes on neural and physiological regulation (Abdulai-Saiku et al., 2025; McCarthy & Arnold, 2011).

Age- and weight-related correlations also differed by sex. In females, daily wheel-running rhythms strengthened with age, potentially associated with oestrogen decline (Habermehl et al., 2022; Nelson et al., 1992), as oestrogen is known to influence daily rhythms (Alvord et al., 2022; Nakamura et al., 2008). By contrast,

5 day rhythms weakened across behaviours with age in females, while no such effects were observed in males. These findings highlight the complexity of ageing-related changes in daily and infradian rhythms, particularly in females, and underscore the need for further studies to explore underlying mechanisms.

### Body-weight regulation following transfer to single housing is shaped by both age and sex

Interestingly, middle-aged female mice showed significant weight loss after transfer to single housing, which correlated negatively with daily locomotor rhythms and with infradian (5 day) rhythms in locomotor, eating and drinking behaviours. Stress from isolation or adaptation to new conditions may contribute, consistent with reported sex differences in stress responsivity in rodents (Bangasser et al., 2010; Curtis et al., 2006) and humans (Bangasser & Valentino, 2014; Kendler et al., 1995). Social isolation can also alter metabolism (Bove et al., 2022), stress hormones (Benfato et al., 2022) and activity, leading to reduced intake or increased energy expenditure (Yamada et al., 2015). Peripheral (ghrelin, leptin) and central (ghrelin) signals acting on hypothalamic and dopaminergic neurons may further underlie sex- and age-dependent feeding and locomotor regulation (Yamada et al., 2015). Increased activity in middle-aged animals, particularly those previously sedentary or overweight in group housing, could also explain weight reduction. Thus, both stress-related and exercise-associated mechanisms likely interact. The negative correlation between body-weight change and daily rhythms suggests that females with stronger daily patterns expended more energy, while stronger 5 day rhythms were linked to less weight loss, indicating a potential role for infradian cycles in energy balance regulation.

### Limitations

A limitation of this study is that we did not monitor body temperature or use indirect calorimetry so we cannot ascertain specific sex- and age-related changes in metabolism underpinning the animals' responses to single housing. Further, we could not assess body-fat deposition, so the tissue-specific origins of the body-weight loss remain unclear. Another limitation is that the robustness of some behaviour rhythms varied over the duration of the experiment and so assessments of rhythm characteristics were specific to particular epochs examined. Finally, we acknowledge that the genetic modification in PER2::LUC mice may limit the generalisation of age- and sex-related changes in daily/circadian and infradian parameters to other mouse strains and rodent species.

Overall, these results highlight the importance of monitoring and examining more than just one behaviour to gain a more extensive understanding of how ageing and sex interact with biological rhythms. Future studies should explore the underlying mechanisms driving these changes, such as hormonal influences, SCN plasticity, and peripheral clock regulation, to better inform interventions aimed at promoting healthy ageing.

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

## Additional information

### Data availability statement

The datasets generated and analysed during the current study are available from the corresponding author on reasonable request.

### Competing interests

The authors declare no competing interests.

### Author contributions

H.D.P., P.C. and T.H. designed the experiments; P.C. and T.H. performed the experiments; C.M., S.L., M.S. and L.C. assisted P.C. and T.H. in running the experiments; P.C. and T.H. analysed the data; H.D.P. directed the study in Bristol; J.M. and V.H.T. provided additional analysis and interpretation. P.C. and H.D.P. wrote the first drafts of the manuscript. All authors contributed to the manuscript.

### Funding

P.C. and T.H. were supported by project grants from the BBSRC to H.D.P. (BB/R019223/1; BB/W000865/1). P.C. was supported by an sLoLa grant from the BBSRC (BB/Z517458/1). C.M. was supported by a BBSRC SWBio Doctoral Training Program studentship (BB/T008741/1), S.L. was supported by a MRC GW4 Doctoral Training Program studentship (MR/W006308/1), and M.S. was supported

by a PhD studentship from the Indonesia Endowment Fund for Education (Ministry of Finance, the Republic of Indonesia). L.C. is supported by a sir Henry Wellcome Research Fellowship (Wellcome Trust, UK; 224116/Z/21/Z). J.M. was financially supported by the National Science and Technology Council (NSTC) (112-2314-B-038-063, 113-2314-B-038-121, 114-2320-B-038-052-MY3), and the Higher Education Sprout Project from the Ministry of Education (MOE) (DP2-TMU-114-N-06) in Taiwan.

## Acknowledgements

The authors thank Professor Emma Robinson for discussion and advice and Dr Chris Marshall for advice and assistance with cytology.

## Keywords

ageing, circadian, drinking, feeding, female, infradian, male, oestrus, sex difference, wheel running

## Supporting information

Additional supporting information can be found online in the Supporting Information section at the end of the HTML view of the article. Supporting information files available:

**Peer Review History**
**Statistical summary**
**Supplementary Figure 1**
**Supplementary Figure 2**
**Supplementary Figure 3**
**Supplementary Figure 4**
**Supplementary Figure 5**
**Supplementary Figure 6**
**Supplementary Figure 7**
**Supplementary Figure 8**
**Supplementary Figure 9**

