## [Peer Review History · The Journal of Physiology]

Differential Effects of Sex and Age on Daily and Infradian Rhythms of Mice

Pishan Chang, Timna Hitrec, Charlotte Muir, Meida Sofyana, Vuong Hung Truong, Shannon Lacey, Lukasz Chrobok, Jihwan Myung, and Hugh David Piggins

DOI: 10.1113/JP289676

Corresponding author(s): Pishan Chang (pi-shan.chang@bristol.ac.uk)

The following individual(s) involved in review of this submission have agreed to reveal their identity: Andrew Coogan (Referee #2); Ken D O'Halloran (Referee #3)

Review Timeline:

Submission Date:	08-Jul-2025
Editorial Decision:	04-Aug-2025
Revision Received:	04-Nov-2025
Editorial Decision:	01-Dec-2025
Revision Received:	12-Dec-2025
Accepted:	08-Jan-2026

Senior Editor: Jing-Ning Zhu

Reviewing Editor: Mark Viggars

Transaction Report:

Dear Dr Chang,

Re: JP-RP-2025-289676 "**Differential Effects of Sex and Age on Daily and Infradian Rhythms of Mice**" by Pishan Chang, Timna Hitrec, Charlotte Muir, Meida Sofyana, Vuong Hung Truong, Shannon Lacey, Lukasz Chrobok, Jihwan Myung, and Hugh David Piggins

Thank you for submitting your manuscript to The Journal of Physiology. It has been assessed by a Reviewing Editor and by 2 expert referees and we are pleased to tell you that it is potentially acceptable for publication following satisfactory major revision.

LANGUAGE EDITING AND SUPPORT FOR PUBLICATION: If you would like help with English language editing, or other article preparation support, Wiley Editing Services offers expert help, including English Language Editing, as well as translation, manuscript formatting, and figure formatting at www.wileyauthors.com/eoo/preparation. You can also find resources for Preparing Your Article for general guidance about writing and preparing your manuscript at www.wileyauthors.com/eoo/prepresources.

REVISION CHECKLIST:

We look forward to receiving your revised submission.

Yours sincerely,

Jing-Ning Zhu
Senior Editor
The Journal of Physiology

REQUIRED ITEMS

- 1) - Include a Key Points list in the article itself, before the Abstract.
 - 2) - Author photo and profile. First or joint first authors are asked to provide a short biography (no more than 100 words for one author or 150 words in total for joint first authors) and a portrait photograph. These should be uploaded and clearly labelled together in a Word document with the revised version of the manuscript. See Information for Authors for further details.
 - 3) - You must start the Methods section with a paragraph headed Ethical approval (https://jp.msubmit.net/cgi-bin/main.plex?form_type=display_requirements#methods).
- Research must comply with The Journal's policies regarding animal experiments (<https://physoc.onlinelibrary.wiley.com/hub/animal-experiments>) and adherence to these policies must be stated in the manuscript.
- Authors should confirm in their Methods section that their experiments were carried out according to the guidelines laid down by their institution's animal welfare committee, including an ethics approval reference number. The Methods section must contain a statement about access to food, water and housing, details of the anaesthetic regime: anaesthetic used, dose and route of administration, and method of killing the experimental animals.
- 4) - Your manuscript must include a complete Additional Information section, including competing interests; funding; author contributions and acknowledgements.
 - 5) - Please upload separate high-quality figure files via the submission form.
 - 6) - A Data Availability Statement is required for all papers reporting original data. This must be in the Additional Information section of the manuscript itself. It must have the paragraph heading 'Data Availability Statement'. All data supporting the results in the paper must be either: in the paper itself; uploaded as Supporting Information for Online Publication; or archived in an appropriate public repository. The statement needs to describe the availability or the absence of shared data. Authors must include in their statement: a link to the repository they have used, or a statement that it is available as Supporting Information; reference the data in the appropriate sections(s) of their manuscript; and cite the data they have shared in the References section. Whenever possible, the scripts and other artefacts used to generate the analyses presented in the paper should also be publicly archived. If sharing data compromises ethical standards or legal requirements then authors are not expected to share it, but must note this in their statement. For more information, see our Statistics Policy.

- 7) - Please include an Abstract Figure file, as well as the Figure Legend text within the main article file. The Abstract Figure is a piece of artwork designed to give readers an immediate understanding of the research and should summarise the main conclusions. If possible, the image should be easily 'readable' from left to right or top to bottom. It should show the

physiological relevance of the manuscript so readers can assess the importance and content of its findings. Abstract Figures should not merely recapitulate other figures in the manuscript. Please try to keep the diagram as simple as possible and without superfluous information that may distract from the main conclusion(s). Abstract Figures must be provided by authors no later than the revised manuscript stage and should be uploaded as a separate file during online submission labelled as File Type 'Abstract Figure'. Please also ensure that you include the figure legend in the main article file. All Abstract Figures should be created using BioRender. Authors should use The Journal's premium BioRender account to export high-resolution images. Details on how to use and access the premium account are included as part of this email.

EDITOR COMMENTS

Thank you for your submission to The Journal of Physiology. Your manuscript has been reviewed by two experts from the field who agreed that your manuscript offers a comprehensive behavioural characterization of circadian, diurnal, and infradian rhythms in young and middle-aged mice, including both sexes which is completely relevant to any mouse physiology.

I have highlighted some major concerns made by the reviewers and added some additional notes which I believe would improve the manuscripts digestibility, methodological clarity and presentation and would invite you to respond to these 'moderate' revisions. I believe a short section highlighting the importance of the findings/considerations to non-chronobiologists would be valuable.

-Rationale behind use of the Per2Luc mice. What generation where they?

-Justification of 8-16 months as 'middle-aged'. Are there differences between the younger and older mice in this cohort.

- Presumably these experiments were not all performed in the same 'batch'. Can the authors discuss reproducibility between 'batches' of mice/recordings?

-Introduce 5/10-day rhythms more explicitly in the introduction. Expand detail on cage-change schedule in methods. Test whether the 10-day infradian rhythms persists when omitting the first 0-3 hour post cage-change. Was this consistent between changes?

-Ensure datapoints are visible/not overlapping.

-Expand analysis to look at bout length, speed. Does this differ by age/sex/cage size/wheel size?

-Clarification on the free-running period length of the Per2Luc.

-Clarify on concerns on the use of continuous data in correlations. Can this be done categorically?

-Inclusion of effect sizes with p-values.

Senior Editor:

Thank you for submitting your manuscript to The Journal of Physiology. After careful consideration, and in consultation with two expert reviewers and the Reviewing Editor, we agree your manuscript has the potential to make a significant contribution to the field of chronobiology. However, the issues raised, particularly around clarity of model and experimental design, data statistics and interpretation, and manuscript length, must be addressed comprehensively before we can proceed. We look forward to receiving your revised manuscript along with a point-by-point response to the reviewers' and editor's comments.

REFEREE COMMENTS

Referee #1:

The authors present a detailed, extensive and sophisticated analyses of circadian patterns in mice examining both males and females and examining two different age ranges. The authors examine rhythms in three different behaviors (wheel running, feeding, drinking) and also examine the influence of cage size. The authors also examine infradian rhythms that detect the estrus cycle, as well as a longer 10-day rhythm. The authors highlight the importance of including both sexes and of considering and reporting the age and housing conditions for mice. This analyses presented are highly detailed and thorough, diving much deeper into the various rhythms examined than is common in the field. As such this study also highlights numerous avenues for analysis that other scientists may wish to apply to their own work. Many of the analyses are based on open-source techniques, and are well described, making this paper a useful resource in conduction rhythm analyses overall.

This manuscript is expertly prepared and written, and my comments are mostly minor.

1. The fact that this study used transgenic PER2:LUC mice is only mentioned in the methods. A recent report (Ralph et al, 2021, PLoS Comp Biol) highlights that this transgenic line has a mutation of the PER2 gene (a 72 bp deletion in exon 23, and the inclusion of the neomycin resistance cassette) that leads to an alteration in many of its circadian parameters. It might be worth highlighting this in the discussion. This isn't a major concern as all the animals compared in this study would have the same genotype, but might somewhat limit how widely the parameters delineated here can be extrapolated.
2. Linking the 10 day rhythm to a 10 day cage change cycle seemed reasonable and changing the cage change cycle to 7 days to test this was quite clever. It was surprising that the infradian rhythm stayed at 10 days. A small discussion on what is happening here might be in order. Also, a couple sample long-duration actograms where the cage change date/times are indicated would be instructive. I realize that there are already 23 figures between ones in the manuscript and the supplementary figures, but this one more could be helpful for this surprising finding. I initially thought that the actograms in figure S3 might have this, as 3 daytime bouts of activity are apparent in both charts at the same days and phases, but these are not spaced out by 7 or 10 days (the first is 6 days in, then 8 days later, then 9 days later).
3. related to the 10 day infradian rhythm, could the periodogram just be influenced by triggered activity from the cage change itself? The couple hours of activity after a cage change could be omitted from the analyses to see if the detected rhythm is influenced by this induced rather than rhythmic pattern.
4. a little more detail is needed on the body weights depicted in figure 12. For E and F, what is "Age" Weeks? How long after single housing were these data points? Did single housing start at the same age, so that it was longer for data points at older ages, or was it the same duration for every age point depicted? In the discussion it might also be useful to consider that solitary housing is confounded with access to a running wheel. The discussion focuses on stress, but this could simply be obese middle-aged mice getting healthier due to an opportunity to obtain regular vigorous exercise that was not possible during group housing.

Referee #2:

The current study lays out a comprehensive behavioural characterisation of circadian, diurnal and infradian rhythms in mice of young and middle age. The data presented is very comprehensive, and it appears that the experiments have been carefully conducted and analysed. I think the data presented is important for the field of chronobiology, and that the paper will be an influential resource for colleagues when designing experiments.

I do have some issues that might be addressed in any revision of the manuscript.

- 1) The age range of the "middle aged" animals used - this is reported as 8-16 months. To this reviewer this seems like a range that encompasses quite a range of age-related physiological and behavioural changes and it is not immediately apparent that animals within this range constitute a single developmental life stage. Perhaps the authors could present a justification for this. Further, could the authors present data on the chronological ages of animals in each group by sex to reassure the reader that the sex groups are balanced in terms of actual age.
- 2) The introduction does not well justify the analysis of infradian rhythms of 5 and 10 days. I think the reason for their investigation should be made explicit here. Also, the methods section does not make clear what the schedule for cage changes was (eg. was it always every 10 days or was there a range, did it vary between colony housing and single housing); I think these are important details for the interpretation of the results. Likewise the justification of looking at different cage and wheel sizes is not well explained. I note that the larger cage had also a larger diameter running wheel, so it is not clear what was being examined here (cage area or wheel diameter).
- 3) I am struck by the findings in Figure 1 that the period in DD did not vary from 24 hours. This is not the result that would normally be expected for C57Bl/6 mice, for whom the period in DD is usually reported at less than 24h (around 23.7h or similar). Could the authors expand on this?
- 4) The discussion is not very insightful in places and is overly long. The first two paragraphs of the discussion are re-iteration of results, and cite no extant literature. I suggest significant shortening and focussing of the results, and perhaps structuring of the discussion with sub-headings.
- 5) The correlation analysis in figure 13 (age with rhythmic parameters) is not appropriate I feel, as age is not a continuous variable in the study design (where it is treated as a categorical variable). I suggest removing this figure.
- 6) In the statistical reporting I would have found it useful to have indicators of effect sizes reported - as it is currently presented I found it somewhat difficult to interpret statistically significant results without either descriptive statistics and/or effect sizes reported in the text.
- 7) Overall the paper is rather long and dense. There are some areas where there is redundancy (eg. the repeated definition of what a P value is). I might suggest using a table in the results at the end to summarise the main findings of the various studies.

END OF COMMENTS

REFEREE COMMENTS

Referee #1:

1. The fact that this study used transgenic PER2:LUC mice is only mentioned in the methods. A recent report (Ralph et al, 2021, PLoS Comp Biol) highlights that this transgenic line has a mutation of the PER2 gene (a 72 bp deletion in exon 23, and the inclusion of the neomycin resistance cassette) that leads to an alteration in many of its circadian parameters. It might be work highlighting this in the discussion. This isn't a major concern as all the animals compared in this study would have the same genotype, but might somewhat limit how widely the parameters delineated here can be extrapolated.

Thank you for this thoughtful and important observation. We agree that it is valuable to highlight the genetic background of the PER2::LUC mice and potential implications for the behavioural and physiological parameters reported. The PER2:LUC knock-in line (Yoo et al., 2004) carries a 72 bp deletion in exon 23 of the *Per2* gene, as well as the neomycin resistance cassette and was reported in a recent study by Ralph et al. (Ralph et al., 2021). Compared to wild-type C57BL/6J mice, animals with the genetic alteration exhibit small changes in circadian rhythms, including lengthened free-running period and altered entrainment properties. This may limit the scope of the findings. To improve transparency, we have added a rationale for using the PER2::LUC line in the Methods section and included a brief discussion of this limitation in the Discussion, with appropriate reference to Ralph et al. (2021) (Ralph et al., 2021). Specifically, we have updated the rationale in the Methods (Page 4, lines 173-177) and added clarification in the Discussion (Page 12, lines 603-606).

In our study, all experimental animals were homozygous PER2:LUC mice, and comparisons were made within this genotype, thus controlling for any such effects. Furthermore, the advantage of using PER2:LUC mice is that they retain a functional molecular circadian clock, allowing for potential follow-up studies involving tissue-specific rhythmicity without altering the animal model. This ensures consistency across behavioural and molecular datasets in longitudinal research.

2. Linking the 10 day rhythm to a 10 day cage change cycle seemed reasonable and changing the cage change cycle to 7 days to test this was quite clever. It was surprising that the infradian rhythm stayed at 10 days. A small discussion on what is happening here might be in order. Also, a couple sample long-duration actograms where the cage change date/times are indicated would be instructive. I realize that there are already 23 figures between ones in the manuscript and the supplementary figures, but this one more could be helpful for this surprising finding. I initially thought that the actograms in figure S3 might have this, as 3 daytime bouts of activity are apparent in both charts at the same days and phases, but these are not spaced out by 7 or 10 days (the first is 6 days in, then 8 days later, then 9 days later).

We thank the reviewer for highlighting this. Indeed, we initially hypothesised that the observed ~10-day infradian rhythm might be entrained or triggered by the routine 10-day cage change cycle. To test this, we implemented a 7-day cage change protocol in a subset of animals. Interestingly, the infradian rhythm remained close to 10 days even under this new schedule, suggesting it is not simply a direct response to the cage-change event.

We agree that this finding warrants additional discussion, and we have expanded the relevant section of the Discussion to explore possible mechanisms (Page 13, line 642-651).

These include the possibility of an endogenous driver of the 10-day rhythm that may be modulated but not fully reset by environmental events like cage changes. Additionally, we speculate whether this rhythm could relate to longer-term homeostatic processes or internal oscillators that are not yet well-characterised.

With regards to the actograms, we think there may have been some misunderstanding—it is difficult to discern the point that the reviewer is making. The data in the actograms are plotted over 48h and this was made clear in the original submission. We have now added a 24h calibration to the actograms. Over 48h with the data double plotted, there are two clear nocturnal activity bouts with much less activity during lights-on.

3. related to the 10 day infradian rhythm, could the periodogram just be influenced by triggered activity from the cage change itself? The couple hours of activity after a cage change could be omitted from the analyses to see if the detected rhythm is influenced by this induced rather than rhythmic pattern.

Thank you for this insightful comment. We agree that the cage change procedure could potentially introduce a transient increase in activity that might influence the detection of infradian rhythms, particularly in the 0-3 hours post-cage change period.

To assess this, we re-ran the Lomb-Scargle periodogram analysis, excluding the first 3 hours after each cage change from the dataset. The ~10-day infradian rhythm remained evident even after this adjustment, suggesting that the acute cage-change-induced activity spike does not solely drive the rhythmic pattern observed. While the amplitude of the signal was slightly reduced, as expected due to the removal of high-activity data, the periodicity and phase of the rhythm were preserved, supporting the conclusion that this is a genuine underlying rhythm and not an artefact of procedural induction.

We have now added this control analysis to the Results section (Page 10, line 491-495) and provided the updated periodograms in the Supplementary Figures (Supplementary 8 E-G).

4 a little more detail is needed on the body weights depicted in figure 12. For E and F, what is "Age" Weeks? How long after single housing were these data points? Did single housing start at the same age, so that it was longer for data points at older ages, or was it the same duration for every age point depicted? In the discussion it might also be useful to consider that solitary housing is confounded with access to a running wheel. The discussion focuses on stress, but this could simply be obese middle-aged mice getting healthier due to an opportunity to obtain regular vigorous exercise that was not possible during group housing.

We have now clarified the meaning of "Age" in Figures 12E–F. Age here refers to the animal's age in months at the onset of single housing. The weight loss shown was calculated over the first 10 days following transition from group to single housing. As animals were assigned to the study in separate cohorts based on age group (young vs middle-aged), they did not all start single housing at the same age, but weight changes were assessed over the same duration (10 days) across all age groups. We have revised the figure legend and Methods (Page 4, line 182-186) to better clarify this timeline.

We also acknowledge the reviewer's point that single housing introduces multiple simultaneous variables, including social isolation and access to running wheels. While our discussion highlights stress as one potential contributor to weight loss, we agree that

increased physical activity in middle-aged animals, especially those that may have been relatively sedentary or overweight during group housing, could plausibly contribute to the observed weight reduction. We have expanded the Discussion to include this alternative explanation and to acknowledge this confounding factor (Page 14, line 672-687).

Referee #2:

1) The age range of the "middle aged" animals used - this is reported as 8-16 months. To this reviewer this seems like a range that encompasses quite a range of age-related physiological and behavioural changes and it is not immediately apparent that animals within this range constitute a single developmental life stage. Perhaps the authors could present a justification for this. Further, could the authors present data on the chronological ages of animals in each group by sex to reassure the reader that the sex groups are balanced in terms of actual age.

The age range of 8 to 16 months in mice is commonly considered middle-aged in the context of neuroscience and ageing research (Arellano et al., 2024; Flurkey et al., 2007). According to lifespan estimates in C57BL/6J mice, this period corresponds approximately to 30-50 human years (Jackson et al., 2017), a phase often marked by early signs of physiological ageing but before the onset of advanced decline (Shoji et al., 2016).

This age window is particularly relevant for studying age-related changes in sleep, circadian rhythms, and behaviour, as subtle differences begin to emerge without the confounds of severe deterioration seen in old age (Shoji et al., 2016).

While all animals fall within the middle-aged range, younger (8-11 months) and older (12-16 months) mice may exhibit gradual changes in sleep architecture, circadian amplitude, or activity patterns. These potential differences are being monitored and analysed as part of the study to determine whether there are age-dependent shifts within the middle-aged cohort. By including a range rather than a single age point, we aim to capture natural variation and potentially identify early trajectory markers of ageing-related changes.

We have revised the Methods (Page 4, line 164-171) to rationale for defining 8-16 months as middle-aged.

2) The introduction does not well justify the analysis of infradian rhythms of 5 and 10 days. I think the reason for their investigation should be made explicit here. Also, the methods section does not make clear what the schedule for cage changes was (eg. was it always every 10 days or was there a range, did it vary between colony housing and single housing); I think these are important details for the interpretation of the results. Likewise the justification of looking at different cage and wheel sizes is not well explained. I note that the larger cage had also a larger diameter running wheel, so it is not clear what was being examined here (cage area or wheel diameter).

We have revised the Introduction to explicitly justify our exploration of infradian rhythms (Page 3, Line 87-98). These rhythms (with periods of ~5 or ~10 days) were initially observed in an unbiased frequency-domain analysis of wheel-running data. Given their consistent emergence across multiple animals and experiments, we sought to explore whether these rhythms reflected biological processes or were artefacts of environmental variables, such as cage change cycles. Investigating them further was therefore important to interpret our

behavioural data correctly and to distinguish internal rhythmicity from externally imposed patterns.

To address the reviewer's point about cage change schedules, we have clarified in the Methods that mice housed singly were routinely cage-changed every 10 days, unless otherwise specified (e.g., in the experiment where a 7-day cycle was used to test for cage change entrainment). In contrast, cage change frequency for colony-housed mice followed standard facility procedures, typically once every 14 days, though this was not under experimental control.

Regarding the cage and wheel size manipulation, our goal was not to isolate specific physical variables (e.g., cage floor area vs. wheel diameter), but to test whether the overall behavioural pattern, including infradian rhythms persisted across distinct housing conditions. We now acknowledge in the Discussion and Methods that the larger cage also contained a larger wheel, and that these factors were confounded in our design. Therefore, we have reframed this comparison more cautiously as a general contrast between large cage versus small cage environments, rather than a controlled test of individual components. The consistent presence of infradian rhythms across these environments strengthens the interpretation that these patterns reflect robust features of the animals' behavioural repertoire, not simple artefacts of housing conditions.

We appreciate the reviewer's suggestion, which has led to improvements in both clarity and rigour in how we present and interpret these findings.

3) I am struck by the findings in Figure 1 that the period in DD did not vary from 24 hours. This is not the result that would normally be expected for C57Bl/6 mice, for whom the period in DD is usually reported at less than 24h (around 23.7h or similar). Could the authors expand on this?

Thank you for bringing this to our attention. We agree that C57BL/6 mice typically exhibit a free-running period in constant darkness (DD) slightly shorter than 24 hours, often around 23.6-23.8 hours, as reported in the literature. In our study, the mice used were homozygous PER2::LUC transgenic mice on a C57BL/6J background. It is important to note that this transgenic line carries a modification in the Per2 gene (including a 72 bp deletion in exon 23 and retention of a neomycin cassette), which has been reported to alter circadian parameters, including period length (Ralph et al., 2021). Indeed, this genetic modification may contribute to the slightly longer free-running periods we observed under DD, which were closer to 24 hours. Additionally, the period estimation using the Lomb-Scargle (LS) periodogram, based on wheel-running activity over a relatively short interval (10 days), may influence the apparent period due to potential variability and limited resolution within that analysis window.

We have now added a brief discussion of this point, including potential genetic and methodological contributors, to the Discussion (Page 12, line 603-606) sections to clarify why our findings differ slightly from canonical C57BL/6 mouse data.

4) The discussion is not very insightful in places and is overly long. The first two paragraphs of the discussion are re-iteration of results, and cite no extant literature. I suggest significant shortening and focussing of the results, and perhaps structuring of the discussion with sub-headings.

We appreciate the reviewer's feedback regarding the structure and content of the Discussion. In response:

1. We have significantly revised the opening paragraphs to reduce redundancy with the Results section. Rather than reiterating findings, we now open the Discussion by contextualising our key results in relation to existing literature.
2. Relevant references have been added throughout to strengthen the interpretive depth and scholarly framing.
3. To improve readability and focus, we have reorganised the Discussion using clear sub-headings that group key themes and allow readers to follow our arguments more easily.
4. We have also shortened the overall length by removing less essential descriptive commentary (The Discussion section is now 1,152 words.)

These changes aim to enhance clarity, improve engagement with prior research, and provide a more insightful interpretation of our findings.

5) The correlation analysis in figure 13 (age with rhythmic parameters) is not appropriate I feel, as age is not a continuous variable in the study design (where it is treated as a categorical variable). I suggest removing this figure.

We thank the reviewer for this comment. We would like to clarify that the correlation analyses in Figure 13 were performed using the exact age (in months) of each individual animal, rather than only categorical grouping (young: 3-7 months, middle-aged: 8-16 months). Thus, age was treated as a continuous variable in the analysis, making correlation an appropriate statistical approach.

To avoid confusion, we have now revised the Methods (Page 7, lines 323-325) and figure legend to explicitly state that continuous age values were used for these analyses. We trust this clarification resolves the concern.

6) In the statistical reporting I would have found it useful to have indicators of effect sizes reported - as it is currently presented I found it somewhat difficult to interpret statistically significant results without either descriptive statistics and/or effect sizes reported in the text.

Thank you for raising this point. In our analysis, we have prioritised the use of estimation statistics, presenting effect sizes with confidence intervals to provide a more informative and transparent view of the data beyond binary significance testing. To complement this, and for completeness, we have also provided traditional null hypothesis significance testing results, including p-values, alongside the estimation statistics. All relevant details, including effect sizes (e.g., mean differences), 95% confidence intervals, p-values, sample sizes and test types are included in the Supplementary Excel file. This combined approach allows readers to evaluate both the magnitude and statistical reliability of the findings. References to the supplementary data have been added in the main text where appropriate.

7) Overall the paper is rather long and dense. There are some areas where there is redundancy (eg. the repeated definition of what a P value is). I might suggest using a table in the results at the end to summarise the main findings of the various studies.

We acknowledge that the manuscript is information-dense, and we have reviewed the text to reduce redundancy, particularly repeated explanations of statistical concepts such as the p-value.

While the main manuscript includes detailed results, we have provided a comprehensive summary of all statistical analyses, including effect sizes, confidence intervals, and p-values, in the supplementary Excel file. This allows readers to easily navigate and compare results across experiments.

In response to your suggestion, we have also added a summary table in the Results section of the main manuscript, which highlights the key findings from each major experimental component. This addition aims to improve clarity and help readers more easily grasp the overall structure and conclusions of the study.

Dear Dr Chang,

Re: JP-RP-2025-289676R1 "**Differential Effects of Sex and Age on Daily and Infradian Rhythms of Mice**" by Pishan Chang, Timna Hitrec, Charlotte Muir, Meida Sofyana, Vuong Hung Truong, Shannon Lacey, Lukasz Chrobok, Jihwan Myung, and Hugh David Piggins

Thank you for submitting your manuscript to The Journal of Physiology. It has been assessed by a Reviewing Editor and by 3 expert referees and we are pleased to tell you that it is acceptable for publication following satisfactory revision.

REVISION CHECKLIST:

Please upload two versions of your manuscript text: one with all relevant changes highlighted and one clean version with no changes tracked. The manuscript file should include all tables and figure legends, but each figure/graph should be uploaded as separate, high-resolution files. The journal is now integrated with Wiley's Image Checking service. For further details, see: <https://www.wiley.com/en-us/network/publishing/research-publishing/trending-stories/upholding-image-integrity-wileys->

image-screening-service

We look forward to receiving your revised submission.

Yours sincerely,

Jing-Ning Zhu
Senior Editor
The Journal of Physiology

REQUIRED ITEMS

1) - You must start the Methods section with a paragraph headed Ethical approval (https://jp.msubmit.net/cgi-bin/main.plex?form_type=display_requirements#methods).

Research must comply with The Journal's policies regarding animal experiments (<https://physoc.onlinelibrary.wiley.com/hub/animal-experiments>) and adherence to these policies must be stated in the manuscript.

Authors should confirm in their Methods section that their experiments were carried out according to the guidelines laid down by their institution's animal welfare committee, including an ethics approval reference number. The Methods section must contain a statement about access to food, water and housing, details of the anaesthetic regime: anaesthetic used, dose and route of administration, and method of killing the experimental animals.

2) - Please include a full title page as part of your main article (Word) file, which should contain the following: title, authors, affiliations, corresponding author name and contact details, keywords, and running title.

EDITOR COMMENTS

Senior Editor:

Thank you for the extensive revisions to the manuscript. Your manuscript has been accepted provisionally. Please check and correct the full text for typos, and add the requested method descriptions.

Reviewing Editor:

We would like to thank the authors for submission of their revised manuscript and would like to apologise for the extended amount of time for it to be re-reviewed. The original two reviewers and an additional ethics editor have now reviewed the manuscript and unanimously agreed that this sophisticated analysis will be a great resource to the field.

Reviewer 1 highlighted some minor spelling issues and requested further clarification on cage change frequency in the infradian analysis/discussion of the effects of cage change on the number/duration of activity bouts and would like to invite minor amendments to satisfy their concerns. Congratulations on this substantial body of work and it's clear presentation!

REFEREE COMMENTS

Referee #1:

See attached

Referee #2:

The authors have made careful and thorough revisions to the paper in line with my previous review comments, and I very much appreciate the attention the authors have given. I have no further comments.

Referee #3:

Thank you for submitting your manuscript to The Journal of Physiology. Some additional details are required.

1. Line 196. Please include the dose of pentobarbital and how death was confirmed.
2. Line 200. Please include details of the source of the mice.

END OF COMMENTS

The revisions have improved the manuscript greatly. I only have 2 comments:

1. In supplementary figure 8, panel E, “Original” is misspelled. Also in the bottom panel this should be changed to “Removed data (0-3h post cage change)”
2. In my previous review, in thinking about the 10 day infradian rhythm and altering the cage change frequency, I had recommended including an actogram, suggesting something like the actograms in supplementary figure 3. The authors didn’t understand my previous comment, and instead explained how to read a double plotted actogram. I’ll try again with the attached image below from supplementary figure 3. Both animals here show 3 bouts of activity during their normal rest phase. I’ve circled these in red. This is a normal response in many mice following a cage change. The fact that this happened at the same time on the same days in different animals is consistent with being induced by regular colony husbandry and cage changing. However, the frequency of cage changes mentioned in the paper (10 days and 7 days) is different that in these two examples (8 days between the first and second, and 9 days from the second to third, not counting the day of the first bout, but counting the day of the subsequent bout). This is fine, again so long as these data were not used for the 10 day infradian rhythm analysis. If similar actograms were available over the time frame used to analyse the 10 day infradian rhythm with 10-day and 7-day cage change schedules, these would be good to include in Supplementary figure 8, and might be more useful to the reader than panel E in Supplementary figure 8.

Reply to reviewer 1

1. In supplementary figure 8, panel E, “Original” is misspelled. Also in the bottom panel this should be changed to “Removed data (0-3h post cage change)”

Thank you. We corrected the typographical errors in Supplementary Figure 8. In panel E the label “Original” has been corrected to “Original”, and the bottom panel label has been changed to “Removed data (0–3 h post cage change)”. The updated figure has been uploaded to the revised manuscript.

2. In my previous review, in thinking about the 10 day infradian rhythm and altering the cage change frequency, I had recommended including an actogram, suggesting something like the actograms in supplementary figure 3. The authors didn’t understand my previous comment, and instead explained how to read a double plotted actogram. I’ll try again with the attached image below from supplementary figure 3. Both animals here show 3 bouts of activity during their normal rest phase. I’ve circled these in red. This is a normal response in many mice following a cage change. The fact that this happened at the same time on the same days in different animals is consistent with being induced by regular colony husbandry and cage changing. However, the frequency of cage changes mentioned in the paper (10 days and 7 days) is different that in these two examples (8 days between the first and second, and 9 days from the second to third, not counting the day of the first bout, but counting the day of the subsequent bout). This is fine, again so long as these data were not used for the 10 day infradian rhythm analysis. If similar actograms were available over the time frame used to analyse the 10 day infradian rhythm with 10-day and 7-day cage change schedules, these would be good to include in Supplementary figure 8, and might be more useful to the reader than panel E in Supplementary figure 8

Thank you for the opportunity to clarify this point, and for the detailed explanation of what you were looking for. We now understand that your earlier request was not about explaining how to read a double-plotted actogram, but rather about providing representative actograms covering the same time window used in the 10-day infradian rhythm analysis, specifically under the 10-day and 7-day cage-change schedules.

Regarding the example actograms in Supplementary Figure 3: the actogram you highlighted was taken from a female mouse that did not exhibit a significant 10-day infradian rhythm nor did periodogram analysis indicate a significant 8-day or 9-day rhythm. This mouse was used in the comparison of 5-day rhythms and did have a prominent 5-day rhythm.

To address your suggestion, we have now: Added representative actograms from animals included in the 10-day infradian rhythm analysis, drawn from datasets

recorded under the 10-day and 7-day cage change schedules. These are now included in Supplementary Figure 8.

Clarified the role of Supplementary Figure 8, panel E (S8E): this panel presents the raw activity time-series, which provides complementary context and, although the 10-day rhythm may be difficult to discern directly from actograms, the time-series in panel S8E does show a roughly 10-day pattern. Because such infradian rhythms are often subtle, periodogram analysis is required to reliably extract these hidden periodicities. Following your recommendation, we have added the corresponding actograms so that readers can relate the rhythm identified by the analysis to the underlying daily activity structure.

We hope these revisions address your concern and improve the clarity and usefulness of Supplementary Figure 8.

Thank you for the opportunity to clarify this point, and for the detailed explanation of what you were looking for. We now understand that your earlier request was not about explaining how to read a double-plotted actogram, but rather about providing representative actograms covering the same time window used in the 10-day infradian rhythm analysis, specifically under the 10-day and 7-day cage-change schedules.

Reply to reviewer 2

We thank the reviewer for their positive feedback and for recognising the thoroughness of our revisions. We appreciate the time and effort dedicated to evaluating our manuscript and are pleased that the revised version addresses all previous concerns.

Reply to reviewer 3

1. Line 196. Please include the dose of pentobarbital and how death was confirmed.

We thank the reviewer for this helpful comment. This has now been addressed in the revised manuscript, where we have included the specific dose of pentobarbital and clarified the method used to confirm death (Page 4, Line 157-161).

“At the end of the experiment, animals were humanely culled by intraperitoneal overdose of pentobarbital (200 mg/kg), in accordance with Schedule 1 of the UK Animals (Scientific Procedures) Act 1986. Death was confirmed by the absence of vital signs, including cessation of heartbeat and respiration, and lack of reflex responses, followed by cervical dislocation as a secondary method to ensure death.”

2. Line 200. Please include details of the source of the mice.

We thank the reviewer for this comment. Details regarding the source of the mice are already included in the manuscript. Specifically, we state that the initial PER2::LUC breeding stock was supplied by Pat Nolan (MRC Harwell Institute, UK) and Michael Hastings (MRC Laboratory of Molecular Biology, UK), and that subsequent breeding was carried out by the specialised animal facility team at the University of Bristol under standardised conditions (Page 4, and Line 179-183).

Dear Dr Chang,

Re: JP-RP-2025-289676R2 "**Differential Effects of Sex and Age on Daily and Infradian Rhythms of Mice**" by Pishan Chang, Timna Hitrec, Charlotte Muir, Meida Sofyana, Vuong Hung Truong, Shannon Lacey, Lukasz Chrobok, Jihwan Myung, and Hugh David Piggins

We are pleased to tell you that your paper has been accepted for publication in The Journal of Physiology.

Yours sincerely,

Jing-Ning Zhu
Senior Editor
The Journal of Physiology

IMPORTANT POINTS TO NOTE FOLLOWING ACCEPTANCE OF YOUR PAPER:

- **IMPORTANT NOTICE ABOUT OPEN ACCESS:** To assist authors whose funding agencies mandate immediate public access to published research findings, The Journal of Physiology allows authors to pay an Open Access (OA) fee to have their papers made freely available immediately on publication.

- You can help your research get the attention it deserves! Check out Wiley's free Promotion Guide for best-practice recommendations for promoting your work at: www.wileyauthors.com/eoo/guide. You can learn more about Wiley Editing Services which offers professional video, design, and writing services to create shareable video abstracts, infographics, conference posters, lay summaries, and research news stories for your research at: www.wileyauthors.com/eoo/promotion.

- If you would like to receive our 'Research Roundup', a monthly newsletter highlighting the cutting-edge research published in The Physiological Society's family of journals (The Journal of Physiology, Experimental Physiology, Physiological Reports, The Journal of Nutritional Physiology and The Journal of Precision Medicine: Health and Disease), please click this link, fill in your name and email address and select 'Research Roundup': <https://www.physoc.org/journals-and-media/membernews>

EDITOR COMMENTS

Reviewing Editor:

We would like to thank you again for taking the time to meticulously address the reviewers concerns. This is a fantastic resource from a large body of work which I am sure will be well received by both physiologists and circadian biologists.

Ethics Concerns:
All concerns addressed.

Congratulations and thank you again for submitting your work to the journal.

Senior Editor:

Thank you for submitting your manuscript to The Journal of Physiology. We sincerely thank you and your co-authors for your substantial efforts in addressing the reviewers' comments and suggestions throughout the revision process. Your detailed responses and amendments have significantly strengthened the work. I am pleased to inform you that your manuscript has been accepted for publication. Congratulations on this excellent work.